# An *in vitro* tumor recurrence model based on platinum-resistant colon cancer cells as a research tool for studying cancer cell dormancy

**Alisa Morshneva**⬤*, **Olga Gnedina**⬤, **Maria Igotti**

Institute of Cytology, Russian Academy of Sciences, St. Petersburg, Russia

\* amorshneva@incras.ru

## Abstract

Modeling cancer recurrence and associated conditions such as cancer cell dormancy remains a significant challenge in cancer research. We developed a novel *in vitro* model based on platinum-resistant tumor cells treated with a platinum drug, in which cells progress through all stages of recurrence within 30–40 days post-exposure. Our results demonstrate that treatment of platinum-resistant colorectal cancer cells with oxaliplatin enriches the population with quiescent, dormant cells arrested in the G0/G1 phase. These cells exhibit increased autophagy, elevated expression of stem cell markers, reduced reactive oxygen species (ROS) levels, and heightened resistance to chemotherapy. A key advantage of our model is the high survival rate of residual cells, enabling the maintenance of a large cell population. This facilitates the application of conventional research techniques that are often limited in other recurrence models due to the small size of residual populations. Overall, this model provides opportunities for a wide range of further research for studying specific recurrence- and dormancy-associated signaling pathways and for identifying novel therapeutic targets for preventing tumor recurrence.

## Introduction

Tumor recurrence remains one of the most pressing challenges complicating cancer treatment. The major cause of late recurrence is the reactivation of residual cancer cells, leading to a new burst of tumor growth that is more difficult to suppress. These residual cancer cells constitute a heterogeneous population, including drug-resistant, dormant, stem and other cell types that survive after cancer treatment [1]. To date, cancer cell dormancy is considered one of the key steps in the development of tumor recurrence [2].

Cancer cell dormancy is a state in which cancer cells reversibly stop proliferation and enter a quiescent phase [3]. These cells remain viable but do not proliferate, which makes them particularly difficult to detect and treat using traditional

**Data availability statement:** All relevant data are within the manuscript and its Supporting Information files.

**Funding:** This study was supported by the Russian Science Foundation (project No. 24-25-20164, https://rscf.ru/project/24-25-20164/) and the St. Petersburg Science Foundation (Russia). The funders had no role in study design, data collection and analysis, decision to publish, or preparation of the manuscript.

**Competing interests:** The authors have declared that no competing interests exist.

approaches targeting proliferating cancer cells. Dormant cancer cells can persist in the body for extended periods — sometimes years or even decades — before potentially reactivating and causing cancer recurrence or metastasis [4].

For a number of reasons, all of these residual cell populations, as well as the overall concept of cancer recurrence, are highly challenging to study [5]. Accordingly, there is a particular need to develop new research models. In response to this need, a number of in vitro models of tumor recurrence have been described in the literature [6,7].

In fact, modeling metastasis, minimal residual disease (MRD), cancer recurrence and cancer dormancy *in vitro* are closely intertwined and can hardly be reviewed separately. These processes represent different parts of the same problem of secondary tumor formation; thus, from this perspective, all these models can be considered the models of tumor recurrence. Consequently, modeling tumor recurrence is an essential tool for both fundamental studies and the search for novel therapeutic approaches to treat recurrent tumors.

*In vitro* tumor recurrence models have been described using both 2D and 3D cell cultures, including monocultures and co-culture systems that reproduce the tumor microenvironment. These models differ not only in their cultivation techniques but also in the inducers used to trigger cellular quiescence, mimicking the latent phase when tumor progression is temporarily paused.

In this context, the objective of this study is to devise and implement an *in vitro* approach for tumor recurrence modeling that reproduces its key stages and can be used as a research tool to study cellular dormancy and tumor regrowth in cancer.

For this purpose, we used platinum-resistant colon cancer cells HCT116. To reproduce recurrence *in vitro* and enable manipulation of the process, we treated platinum-resistant cells with high doses of a platinum drug. Platinum drugs are widely used in chemotherapeutic cancer treatment [8]. Their antitumor action is based on the formation of platinum-DNA adducts, impairing DNA replication and repair and eventually leading to tumor cell apoptosis [9].

To some extent, such drug-resistant cancer cells already represent a recurrent tumor as a population of cells that survived chemotherapy. Among other reasons, we chose drug-resistant cells for modeling to avoid excessive cell death and enrich the residual population.

In this study, we presented a reproducible *in vitro* model of tumor recurrence, designed to investigate the biological processes underlying cancer recurrence. In our model, platinum-resistant human tumor cells were exposed to high-dose oxaliplatin or cisplatin. While these treatments induced cytotoxic effects in sensitive cells, the resistant population evaded cell death and instead acquired a dormant phenotype. Following drug removal, the dormant cells resumed proliferation within 25–30 days. A key advantage of our approach lies in its capacity to produce dormant tumor cells in quantities large enough to support comprehensive functional studies such as drug screening, CRISPR-based screens, and other applications, marking a significant advancement over existing models.

## Materials & methods

### Cell lines and cell culture

The study was conducted on platinum-resistant cells obtained as described in Morshneva et al., 2022, from HCT116 human colorectal carcinoma cells from the ATCC collection (RRID:CVCL_0291).

HCT116, HCT116 cspl-R and HCT116 oxpl-R cells carrying one of the FUCCI components [10] – the plasmid vector pLL3.7m-Clover-Geminin(1–110)-IRES-mKO2-Cdt1(30–120) RRID:Addgene_83841 (Addgene, 83841) with green (Clover) and red (mKO2) fluorescent labels labelling Geminin and Cdt1, respectively, were obtained by lentiviral transduction. pLL3.7m-Clover-Geminin(1–110)-IRES-mKO2-Cdt1(30–120) was a gift from Michael Lin (Addgene plasmid #83841; http://n2t.net/addgene:83841; RRID:Addgene_83841). Selection of transduced cells from the G2-enriched population (green label) of nocodazole-treated cells (100 ng/ml) was carried out using cell sorting (BioRad S3e Cell sorter, 530/30 nM).

Cells were cultured in high-glucose Dulbecco's modified Eagle's medium (DMEM) (Biolot, Russia) with 10% fetal bovine serum (HyMedia, India) and 40 µg/ml gentamicin. Cells were treated with 25–50 µM cisplatin (cspl) or 50–150 µM oxaliplatin (oxpl) for 6 or 24 hours. For recurrence modeling, we treated each cell line with a corresponding platinum drug: cisplatin for cisplatin-resistant cells (HCT116 cspl-R) and oxaliplatin for oxaliplatin-resistant cells (HCT116 oxpl-R).

During cultivation, platinum-resistant cells were not continuously treated with platinum, as platinum resistance remains stable and persists even after cryopreservation. If a decrease in resistance was detected, cells were briefly re-exposed to platinum compounds to re-establish their resistant phenotype. This approach ensures consistent resistance levels throughout experiments.

### RNA isolation and real-time quantitative PCR (qPCR)

Total RNA was isolated from cells using TRIzol reagent (Invitrogen, USA) following the manufacturer's instructions. Reverse transcription was performed with 2 µg of RNA using the RT kit (Biolabmix, Russia).

The PCR program consisted of 38 cycles of denaturation at 95°C for 30 s, annealing at 60°C for 30 s, and elongation at 72°C for 30 s. Gene expression levels were normalized to GAPDH and relative quantification was performed using the $2^{-\Delta\Delta Ct}$ method (Life technologies).

All oligonucleotides used for qPCR are listed below (F = forward primer, R = reverse primer):

GAPDH F-ACCATCTTCCAGGAGCGAGA, R-GACTCCACGACGTACTCAGC; *p21/Waf1 (CDKN1A)* F-AGGTGGACCTGGAGACTCTCAG, R-TCCTCTTGGAGAAGATCAGCCG; *p16/Ink4* (CDKN2A) F-CTCGTGCTGATGCTACTGAGGA, R-GGTCGGCGCAGTTGGGCTCC; *Cyclin A* (CCNA) F-CTCTACACAGTCACGGGACAAAG, R-CTGTGGTGCTTTGAGGTAGGTC; Cyclin B1 (CCNB1) F-GACCTGTGTCAGGCTTTCTCTG, R-GGTATTTTGGTCTGACTGCTTGC; *Cyclin E* (CCNE) F-CAGATGGAGCTTGTTCAGGAGAT, R-TTCAGCCAGGACACAATAGTCA; *AURKA* F-GCTGGAGAGCTTAAAATTGCAG, R-TTTTGTAGGTCTCTTGGTATGTG; *MKI67* F-GAAAGAGTGGCAACCTGCCTTC, R-GCACCAAGTTTTACTACATCTGCC; *E2F1* F-GACGTGTCAGGACCTTCGTA, R-CAGGAAAACATCGATCGGGC; *E-cadherin (CDH1)* F-GCCTCCTGAAAAGAGAGTGGAAG, R-TGGCAGTGTCTCTCCAAATCCG; *N-cadherin (CDH2)* F-CCTCCAGAGTTTACTGCCATGAC, R-GTAGGATCTCCGCCACTGATTC; *Vimentin (VIM)* F-CTGCCAACCGGAACAATGAC, R-CATTTCACGCATCTGGCGTT; *TWIST1* F-GCCAGGTACATCGACTTCCTCT, R-TCCATCCTCCAGACCGAGAAGG; *OCT4* F-CTGTCTCCGTCACCACTCTG, R-AAACCCTGGCACAAACTCCA; *SOX2* F-GCTACAGCATGATGCAGGACCA, R-TCTGCGAGCTGGTCATGGAGTT; *NANOG* F-CTCCAACATCCTGAACCTCAGC, R-CGTCACACCATTGCTATTCTTCG; *Clusterin (CLU)* F-AGCTGCTAAAGTCCTACCAGTG, R-CACCCAGTTAAACTGCTCGTTC; *CD44* F-AAATGGTCGCTACAGCATCTCT, R-AATCCGATGCTCAGAGCTTTCT; *Beclin (BECN1)* F-CTGGACACTCAGCTCAACGTCA, R-CTCTAGTGCCAGCTCCTTTAGC; *LC3* F-GAGAAGCAGCTTCCTGTTCTGG, R-GTGTCCGTTCACCAACAGGAAG.

 

## Western blotting

For immunoblotting, cells were lysed in a buffer containing 1% NP-40, 0.5% sodium deoxycholate, 0.1% SDS, 20 mM glycerophosphate, 1 mM sodium orthovanadate, 5 mM EGTA, 10 mM sodium fluoride, 1 mM phenylmethylsulfonyl fluoride, and a protease inhibitor cocktail. Proteins were separated by electrophoresis on 8−15% polyacrylamide gels containing 0.1% SDS, transferred onto a PVDF membrane, and probed with the appropriate primary antibodies. The primary antibodies used were against LC3 #4108 RRID: AB_2137703, Beclin #3738 RRID:AB_490837, Survivin #2808 RRID:AB_2063948 (Cell Signaling, USA), Cyclin A sc-751 RRID:AB_631329, and p27 (C-19) sc-528 RRID:AB_632129 (Santa Cruz Biotechnology, USA) with α-Tubulin sc-32293 RRID:AB_628412 (Santa Cruz Biotechnology, USA) serving as a loading control. Anti-mouse (Jackson ImmunoResearch Labs Cat#315-035-003, RRID:AB_2340061) and anti-rabbit (Jackson ImmunoResearch Labs Cat#111-035-003, RRID:AB_2313567) antibodies conjugated with horseradish peroxidase were used as the secondary antibodies. Visualization of membrane-bound proteins was performed by enhanced chemiluminescence (ECL, Amersham Biosciences).

Each protein of interest was analyzed at least three times using different sample sets. Band intensities were quantified using ImageJ (v1.53e), normalized to loading controls, analyzed via GraphPad Prism (v.8.4.3) [11], and is presented in relative units below each band. To calculate p-ERK:p-p38 ratio (%) based on the densitometry data, the sum of tubulin-normalized p-ERK and p-p38 densities was set to 100% at each time point, allowing calculation of the relative ratio between p-ERK and p-p38.

## DCF staining (ROS detection)

For detection of reactive oxygen species (ROS) cells were incubated with 2',7'-dichlorofluorescin diacetate (DCFDA), which is acetylated inside the cells to form a non-fluorescent compound that is subsequently oxidized by ROS to fluorescent 2',7'-dichlorofluorescin (DCF). Cells were washed with warm PBS and incubated for 20 minutes at 37C in 5 μM DCFDA/PBS, washed again in warm PBS and then analyzed using a flow cytometer Coulter Epics XL (Bechman, USA) with an excitation wavelength of 488 nm.

## MTT cell viability test

Cells were seeded in a 96-well plate at a density of 15,000 cells/well in 100 μL of cell culture medium (DMEM, 10% FBS) with compounds to be tested. Cell viability was assessed using 3-(4,5-dimethylthiazol-2-yl)-2,5-diphenyl tetrazolium bromide (MTT) (Sigma, USA), which is converted by intracellular metabolites of living cells into a water-insoluble purple-colored formazan product. Cells were incubated with 0.5 mg/mL MTT solution at 37°C for 1 hour in a $CO_2$ incubator. Then the medium was aspirated, and the formazan granules were dissolved in DMSO. Optical density (OD) was measured at 570 nm using a Multiskan-EX microplate reader (Thermo Fisher Scientific, Waltham, MA, USA), with clear DMSO as the blank.

## Software and statistic analysis

The microscopic and immunoblot images were analyzed using ImageJ (v1.53e) [11]. Statistical analysis and data visualization were performed in RStudio using R version 4.2.2 (RStudio, Boston, MA, USA) and GraphPad Prism 9.5.1 RRID:SCR_002798 (GraphPad Software, Boston, MA, USA). The plots represent the mean of 3 independent experiments±standard error of the mean (SEM) or 3 independent samples±standard deviation (SD), the metric applied is included in the labels for each figure. Results were checked for statistical significance with a Mann–Whitney U test (ns $p > 0.05$, * $p < 0.05$, ** $p < 0.01$).

## Results

### Reproducing the main stages of cancer recurrence in vitro in platinum-resistant colon cancer cells HCT116

One of the major limitations in studying residual populations of cancer cells, particularly, cell dormancy, is the small size of such sub-populations. Working with drug-resistant tumor cells allows us to overcome this limitation, as only a small

 

fraction of resistant cells die upon exposure to cytotoxic treatment, thereby enriching the residual population. Platinum-resistant cells derived from human colon cancer cells HCT116, specifically cisplatin-resistant (HCT116 cspl-R) and oxaliplatin-resistant (HCT116 oxpl-R) were obtained by cyclic exposure of parental HCT116 cells to platinum drugs as described previously [12]. After 4–6 rounds of platinum exposure, both cisplatin- and oxaliplatin-resistant cells exhibited resistance indices (RI) > 10, demonstrating strong drug resistance (S1A,B Fig).

According to our experimental data, cisplatin-resistant HCT116 cspl-R cells do not exhibit cross-resistance to oxaliplatin, a next-generation platinum drug (S1D Fig). However, they show slight cross-resistance to other DNA-damaging agents, including actinomycin D, etoposide, and adriamycin (doxorubicin). Notably, these cells remain sensitive to the topoisomerase I inhibitor irinotecan, the antimetabolite 5-fluorouracil, and histone deacetylase inhibitors such sodium butyrate (data not shown).

Oxaliplatin-resistant cells also demonstrate high resistance to cisplatin, at a level comparable to cisplatin-resistant cells, making them a more versatile model of generalized resistance to platinum drugs (S1D Fig). Nevertheless, the overall spectrum of drugs to which HCT116 oxpl-R and HCT116 cspl-R cells exhibit cross-resistance largely overlaps.

The *in vitro* model of cancer recurrence presented in this study is inspired by the process of acquiring drug resistance itself, since each round of platinum exposure followed by cell recovery can be viewed as an episode of cancer recurrence.

To model the main stages of cancer recurrence, including the quiescent or dormant state, we expose platinum-resistant HCT116 cells to platinum drugs, causing partial cell death between days 2 and 4 after exposure, typically eliminating no more than half of the cell population. The surviving cells arrest cell cycle progression and enter a quiescent or dormant state lasting from day 4 to days 12–18, after which they resume proliferation and completely repopulate. Thus, the recurrence cycle is completed within 30–40 days after platinum exposure. (Fig 1A).

According to our observations, the transition to the quiescent state is accompanied by cell hypertrophy, multinucleation, and prominent cytoplasmic vacuolization (Fig 1B).

To validate our model as a research tool for studying cellular dormancy in cancer and to confirm that this quiescent state approximates the dormant state reported in the literature, we further analyzed the characteristics of these cells.

## Cells in cancer recurrence model enter quiescence (G0)

Dormant cancer cells are known to stop progressing through the cell cycle in the G0/G1 phase [13,14]. To provide real-time visualization of cell cycle distribution in our model, we used one component of the Fluorescent Ubiquitination-Based Cell Cycle Indicators (FUCCI) system [10]. For this end cells were transduced with a plasmid vector, encoding fluorescently labeled cell-cycle markers Geminin (S-G2-M) and Cdt1 (G0/G1), labelled green and red, respectively (Fig 2A).

Recurrence modeling on FUCCI-labeled cells reveals a pronounced accumulation of red fluorescent (mKO2-Cdt1) upon entering the quiescent state, suggesting transition to G0/G1 phase (Fig 2b,c). Cells in G0/G1 are prevalent until day 7–9. After that, individual cells switch to early S-phase, which is marked by colocalized green and red signals, visible as yellow, indicating the onset of cell repopulation. Concurrently, a reduction in the total red signal reflects progressive exit from G0/G1. The cell cycle distribution tends to return to its initial value.

These findings illustrate the temporal progression from chemotherapy-induced arrest (G0/G1 dominance) to proliferative recovery, highlighting the utility of FUCCI markers in tracking cell cycle re-entry and population restoration.

To comprehensively characterize cell cycle regulation during dormancy and recurrence, we analyzed the dynamics of proliferation markers and cyclin-dependent kinase (CDK) inhibitors expression over 33 days following cisplatin treatment in our tumor recurrence model.

qPCR analysis revealed that expression of proliferation markers such as Aurora kinase A (AURKA), MKI67, E2F1, and cyclins A and B significantly decreased 3–6 days after oxaliplatin treatment in both HCT116 oxpl-R and HCT116 cspl-R cell lines (Figs 2D, 3A). This decline coincided with peak mKO2/Cdt1 staining (marking G0/G1 phase) (Fig 2A). Expression level of these markers returns to baseline after repopulation (day 33). Conversely, CDK inhibitors p21/Waf1 and p27/Kip1 accumulate during the quiescent state (days 1–9), followed by downregulation upon cell cycle re-entry (Fig 3B,C).

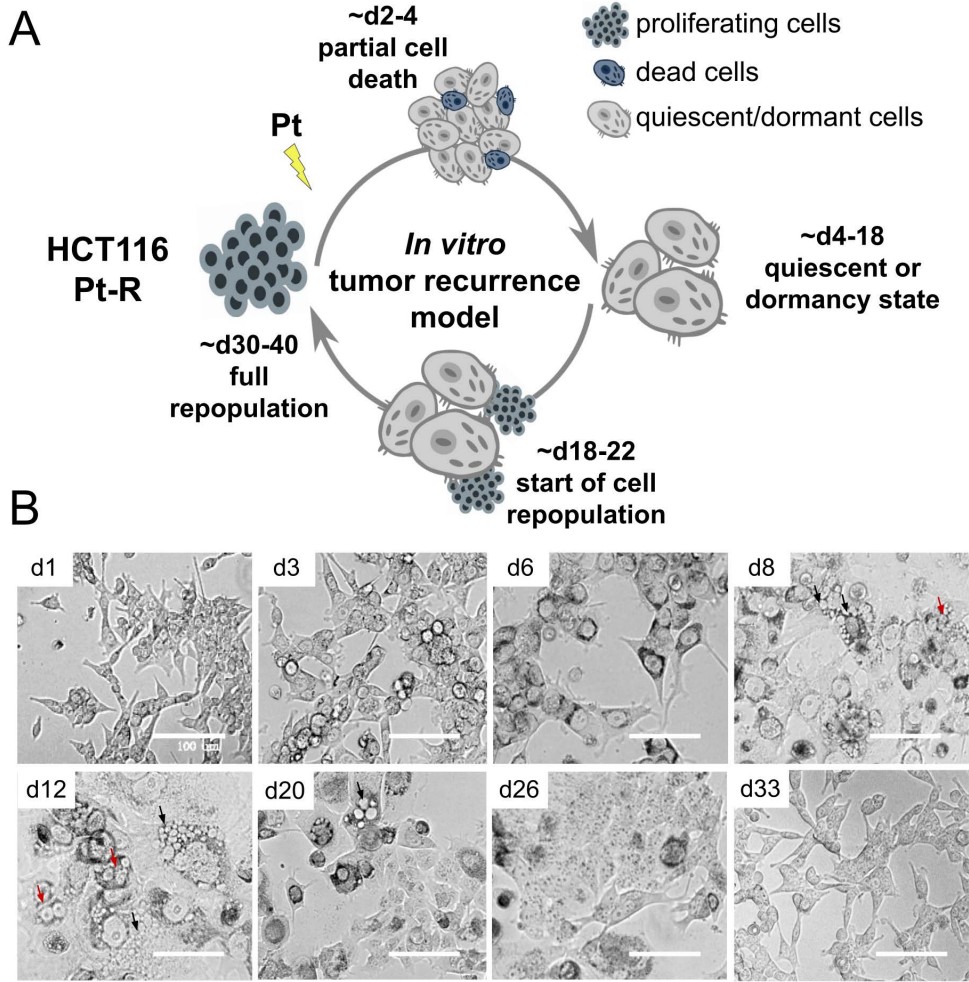

**Fig 1. In vitro cancer recurrence model based on platinum-resistant colon cancer cells HCT116. (A)** Schematic presentation of the model. **(B)** Representative images of cisplatin-resistant cells HCT116 (HCT116 cspl-R) at 1-33 days after exposure to cisplatin. Arrows on the microscopic images indicate areas of vacuolization (black) and multinucleation (red). All images are shown at the same scale; scale bars represent 100 μm. Uncropped images are available in Supplementary S2 Fig.

These molecular data provide independent evidence supporting our initial FUCCI-based conclusions regarding cell cycle arrest and recovery.

Although some authors suggest cellular senescence as a key mechanism in maintaining tumor dormancy [15], our data show no significant accumulation of the senescence marker p16/Ink4 in quiescent cells (Fig 3B), arguing against senescence as the primary dormancy mechanism in our model.

Another hallmark of dormant cancer cells is the divergent change in phosphorylation states of ERK and p38 kinases, with decreased ERK phosphorylation and increased p38 phosphorylation [16]. This pattern is also evident in our tumor recurrence model. Immunoblot analysis of p-p38 and p-ERK levels up to 33 days after cisplatin treatment (Fig 3D) shows accumulation of p-p38 during recurrence modeling accompanied by temporary suppression of p-ERK, resulting in a shifted p-ERK:p-p38 ratio (Fig 3E,F). The most pronounced p-p38 accumulation occurs after day 12, during early repopulation. At complete repopulation, the p-ERK:p-p38 ratio returns to its initial balance.

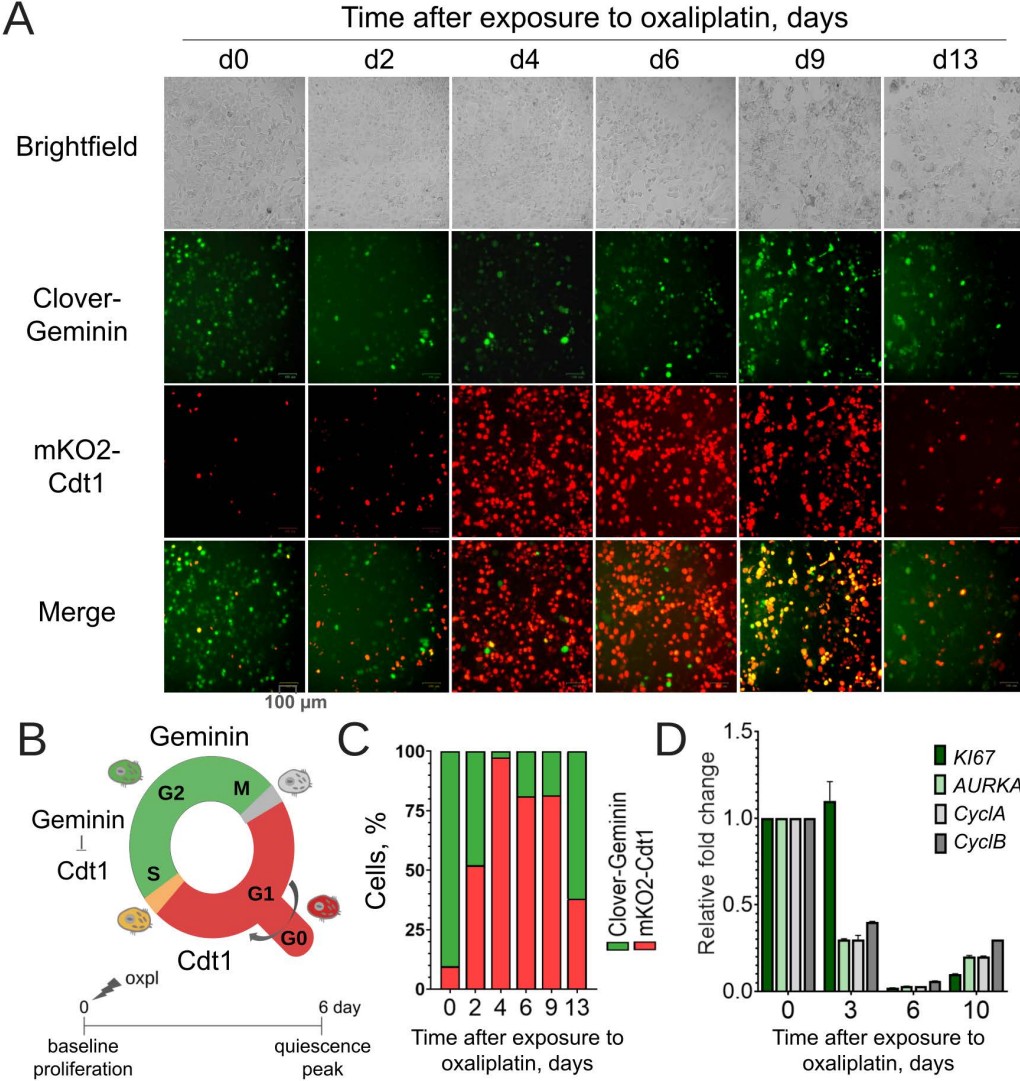

**Fig 2. Cell cycle dynamics and proliferation gene expression in FUCCI-expressing oxaliplatin-resistant HCT116 cells following 150 µM oxaliplatin treatment for 6 hours. (A) Representative fluorescent microscopy images showing FUCCI signals over time.** Red (mKO2-Cdt1) marks G1 phase; green (Clover-Geminin) marks S/G2/M phases. Scale bars: 100 µm. **(B)** Schematic representation of the experimental timeline and cellular redistribution following oxaliplatin treatment. The diagram illustrates the transition of cells from a proliferating state to a maximally enriched population of resting/dormant cells by day 6 post-oxaliplatin treatment. The sequential fluorescence of FUCCI markers is depicted, showing cells in G1 phase (red fluorescence); cells in S/G2/M phases (green fluorescence); cells transitioning between phases (yellow/orange (red+green overlap)). The dynamic shift in fluorescence reflects cell cycle arrest and enrichment of quiescent/dormant cells following chemotherapy-induced stress. **(C)** Quantification of cell cycle phase distribution over time. Bars represent the mean percentage of cells in G0/G1, S, or G2/M phases based on FUCCI signal classification. **(D)** Normalized expression of proliferation-related genes MKI67, AURKA, cyclin A, cyclin B after oxaliplatin exposure (days 0-10), relative to day 0, measured by qPCR, with GAPDH as endogenous control. Bars represent mean relative expression (± SD) from triplicate samples at days 0, 3, 6, and 10 post-treatment.

## Quiescent cells in cancer recurrence model are less sensitive to cytotoxic exposure

Since chemotherapeutic drugs usually target proliferating cells, quiescence and slow proliferation rates in cancer cells are known to be significant limitations that undermine high chemotherapeutic efficacy [17].

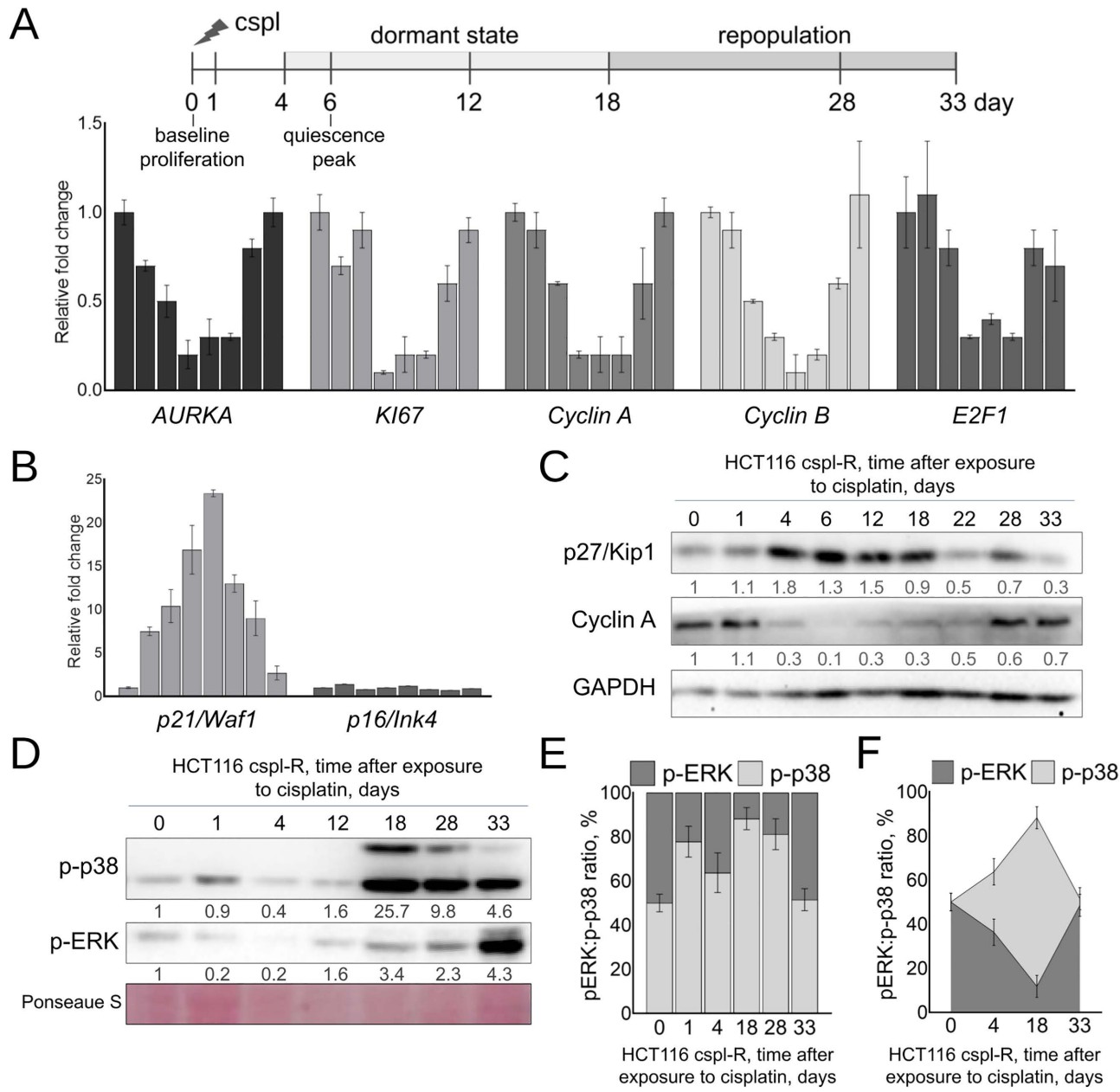

**Fig 3. Dynamics of cell cycle regulators in cancer recurrence model based on cisplatin-resistant colon cancer cells HCT116 cspl-R.** (A,B) qPCR analysis of proliferation markers and CDK inhibitors following cisplatin exposure (days 0-33). Expression normalized to day 0, with GAPDH as endogenous control. Bars represent 0, 1, 4, 6, 12, 18, 28, and 33 days after cisplatin exposure. **(C)** Immunoblot analysis of p27/Kip1 and Cyclin A protein levels (α-Tubulin loading control). Numbers indicate normalized densitometry values relative to day 0. **(D)** Immunoblots with p-p38 and p-ERK antibodies. **(E)** Quantitative p-ERK:p-p38 ratio (%) based on the densitometry data of normalized p-p38 and p-ERK expression (stacked barplot) and (F) schematic representation of signaling balance during dormancy-recurrence transition. All data represent biological triplicates, mean±SEM.

We compared the relative viability of proliferative (day 0, not exposed to oxaliplatin) and quiescent (day 6 after 6-hour exposure to 150 μM oxaliplatin) platinum-resistant colon cancer cells HCT116 oxpl-R after 48 hours of exposure to chemotherapeutic drugs (Fig 4) and proved residual population of quiescent cancer cells to be less sensitive to

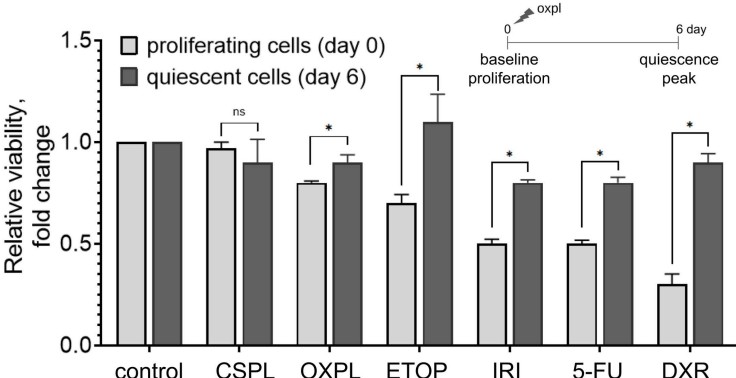

**Fig 4. Relative viability (MTT assay) of proliferating (day 0) and quiescent (day 6 after 6-hour exposure to 150 µM oxaliplatin) platinum-resistant colon cancer cells HCT116 oxpl-R after 48-hour exposure to chemotherapeutic drugs: 25 µM cisplatin (CSPL), 10 µM oxaliplatin (OXPL), 30 µM etoposide (ETOP), 50 µM irinotecan (IRI), 20 µM 5-fluorouracil (5-FU), 0.5 µg/ml doxorubicin (DXR).** Data represent triplicate samples and are displayed as mean±SD.

chemotherapy. Specifically, quiescent cells were completely resistant to etoposide and showed reduced sensitivity to irinotecan, 5-fluorouracil, and doxorubicin used at the IC50 concentration for parental sensitive cells HCT116.

## Quiescent state in cancer recurrence model is associated with high autophagy levels

Autophagy is widely recognized as a cornerstone for the survival of dormant cancer cells [13]. In our tumor recurrence model, quiescent cells exhibit a marked increase in cell size, vacuolization and multinucleation. Multiple vesicles stain positive for acidic lysosomal compartments using LysoTracker™, suggesting they are autophagolysosomes (Fig 5A,B).

To investigate whether quiescent cells in our cancer recurrence model exhibit increased autophagy, we tracked autophagy marker expression (mRNA and protein) for 33 days following cisplatin treatment (Fig 5C,D). Our recurrence model demonstrates biphasic regulation of autophagy. While Beclin-1 levels remain elevated during dormancy (Days 4–14), quantitative analysis reveals a significant decrease during repopulation (Day 21) compared to peak dormancy levels (Fig 5C; S3 Fig), suggesting that partial autophagy downregulation accompanies tumor regrowth. Notably, survivin, a protective protein involved in suppressing excessive autophagy activity [18], accumulates in repopulating cells (days 22–33), coinciding with a decline in LC3 and Beclin levels (Fig 5C). This inverse correlation supports the hypothesis that survivin accumulation suppresses autophagic flux during dormancy.

## EMT in cancer recurrence model in vitro

It is generally accepted that epithelial-mesenchymal transition (EMT), which enables cancer cells to enhance their mobility, invasion, and resistance to apoptotic stimuli, plays an important role in metastasis [19]. Recent findings regarding cancer cell dormancy have expanded the understanding of EMT, showing that its role goes beyond metastasis and invasion and includes involvement in dormancy control [20].

To confirm whether cells in our tumor recurrence model undergo EMT, we analyzed the dynamics of EMT marker expression over 33 days after cisplatin treatment using qPCR (Fig 6A).

It's known that Twist1 activation triggers EMT, resulting in accumulation of N-cadherin and the mesenchymal marker Vimentin along with suppression of E-cadherin [21]. Following cisplatin treatment, we observed accumulation of TWIST1 transcript in quiescent cells, a similar trend was shown for *N-cadherin*.

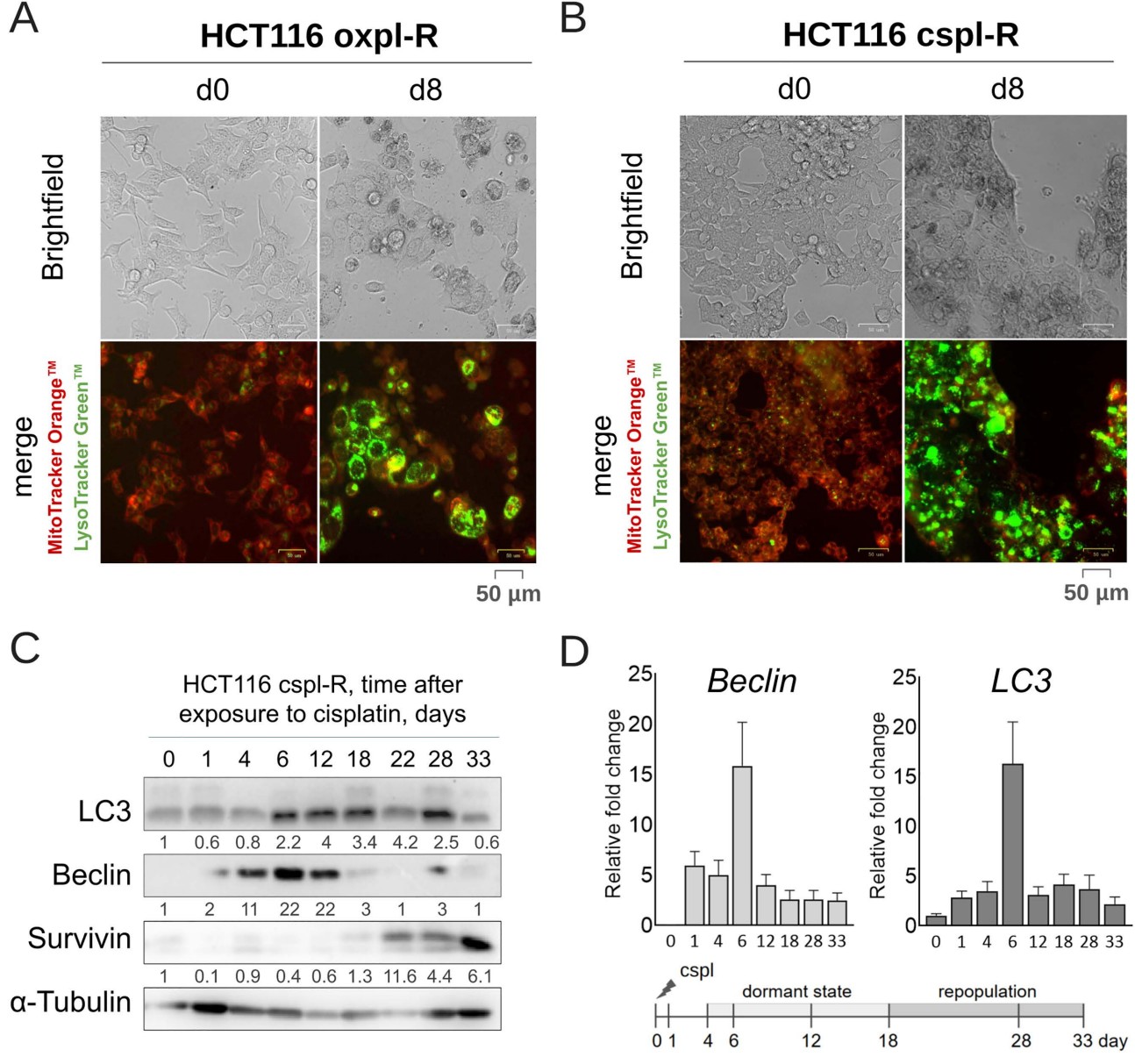

**Fig 5. Autophagy in in vitro cancer recurrence model based on platinum-resistant colon cancer cells. (A,B)** LysoTracker-positive acidic lysosomal compartments in quiescent cells. Oxaliplatin-resistant HCT116 oxpl-R cells (A) and cisplatin-resistant HCT116 cspl-R cells (B) before (day 0) and after (day 8) platinum treatment (oxaliplatin or cisplatin, respectively), stained with LysoTracker Green™ and MitoTracker Orange™. Scale bars represent 50 μm. **(C,D)** Dynamics of autophagy marker expression in the *in vitro* cancer recurrence model based on cisplatin-resistant colon cancer cells HCT116 cspl-R, analyzed by immunoblotting (C) and qPCR **(D)**. **(C)** Immunoblots probed with LC3, Beclin and Survivin antibodies. α-Tubulin was used as a loading control. Quantification values shown beneath each lane represent band intensity normalized to loading control and relative to day 0 control (set as 1.0). **(D)** Normalized expression of autophagy-related genes (*lc3, beclin*) after cisplatin exposure (days 0-33), relative to day 0. Expression of the GAPDH gene served as the endogenous control. Data represent biological triplicate experiments and are displayed as mean±SEM.

Contrary to expectations, no negative correlation was observed between *N-cadherin* (a mesenchymal marker) and *E-cadherin* (an epithelial marker) expression. According to our data, *E-cadherin* also accumulates in quiescent cells (Fig 6A). The reason for this somewhat contradictory result is not entirely clear. We observed the concurrent upregulation of

E-cadherin and N-cadherin, which appears contradictory to the canonical EMT paradigm. While EMT is typically characterized by E-cadherin loss and N-cadherin gain [22,23], accumulating evidence suggests that cells can exhibit partial or hybrid EMT states, wherein both markers are co-expressed [24–26].

However, the accumulation of *Vimentin* after *N-cadherin* activation, as well as the increase in TWIST1 levels, supports the occurence of EMT in our tumor relapse model. We describe the cell phenotype as mesenchymal during days 12–28 days after cisplatin treatment, coinciding with the period of return to active proliferation and repopulation.

## Markers of stemness in cancer recurrence model in vitro

Residual cancer cells are widely viewed through the lens of stemness [27]. Features of dormant cancer cells and cancer stem cells (CSCs) overlap significantly [28], making it difficult to clearly distinguish between dormant cancer cells and quiescent stem cells, especially since dormant cancer cells exhibit enhanced stem-like properties [1].

To explore stemness signatures in our model, we analyzed the expression of stemness-associated markers from 0 to 33 days after cisplatin treatment using qPCR (Fig 6B,C).

The expression of pluripotency-associated stem cell markers (SOX2, OCT4, NANOG) gradually increases and peaks by day 6 after cisplatin treatment (Fig 6B). Similar dynamics were observed for CSC markers *clusterin* and CD44 (Fig 6C). Generally, cells express stemness markers from days 4–18, after which expression declines upon repopulation.

To assess the degree of activation of stemness-related markers in quiescent cells in our model, we compared marker expression at day 6, the peak expression point, with levels in human induced pluripotent stem cells (iPSCs) AD3 (Fig 6D,E). According to qPCR data, OCT4 transcriptional activation in quiescent cells is low compared to iPSCs, while SOX2 and NANOG mRNA levels differ from iPSCs by less than one order of magnitude. In contrast, quiescent cells at day 6 express CSC markers at levels comparable to iPSCs, demonstrating similar clusterin expression and tenfold higher CD44 expression, which is a negative marker for iPSCs [29].

These results suggest that quiescence in our model is accompanied by enhanced stem-like properties.

## Reduced ROS-levels in a non-proliferative state in cancer recurrence model in vitro

It has been established that the dormant or quiescent state is accompanied by reduced levels of reactive oxygen species (ROS) compared to proliferating cells, therefore, redox status is considered a potential marker of dormant cancer cells [30].

We measured ROS levels in HCT116 oxpl-R and HCT116 cspl-R cells in both proliferative and presumably dormant state — corresponding to day 6 of the cancer recurrence model — using flow cytometry of DCF-stained cells (Fig 7).

In oxaliplatin-resistant cells, we showed a significant decrease in ROS levels upon entering dormant or quiescent state (Fig 7A,C,E). In cisplatin-resistant HCT116 cspl-R cells, the baseline ROS level is much lower compared to HCT116 oxpl-R cells, and the decrease in ROS levels during the transition to the dormant state is observed only as a trend (Fig 7B,D,F).

## Quiescent state of cells in the presented model of cancer recurrence can be viewed as dormant

The results described above collectively support the conclusion that the quiescent state of cells in our *in vitro* cancer recurrence model closely resembles the dormant state of cancer cells reported in the literature. Based on a range of characteristics, including expression level of markers linked to proliferation, EMT, stemness, autophagy, as well as ROS levels, platinum-resistant cells can be considered dormant between 4 and 14 days after treatment with a platinum drug. Therefore, our model of cancer recurrence *in vitro* can also be regarded as a model of cellular dormancy in cancer.

This model offers broad opportunities for scientific research on cancer cell dormancy and for the development of novel strategies aimed at eliminating dormant cancer cells.

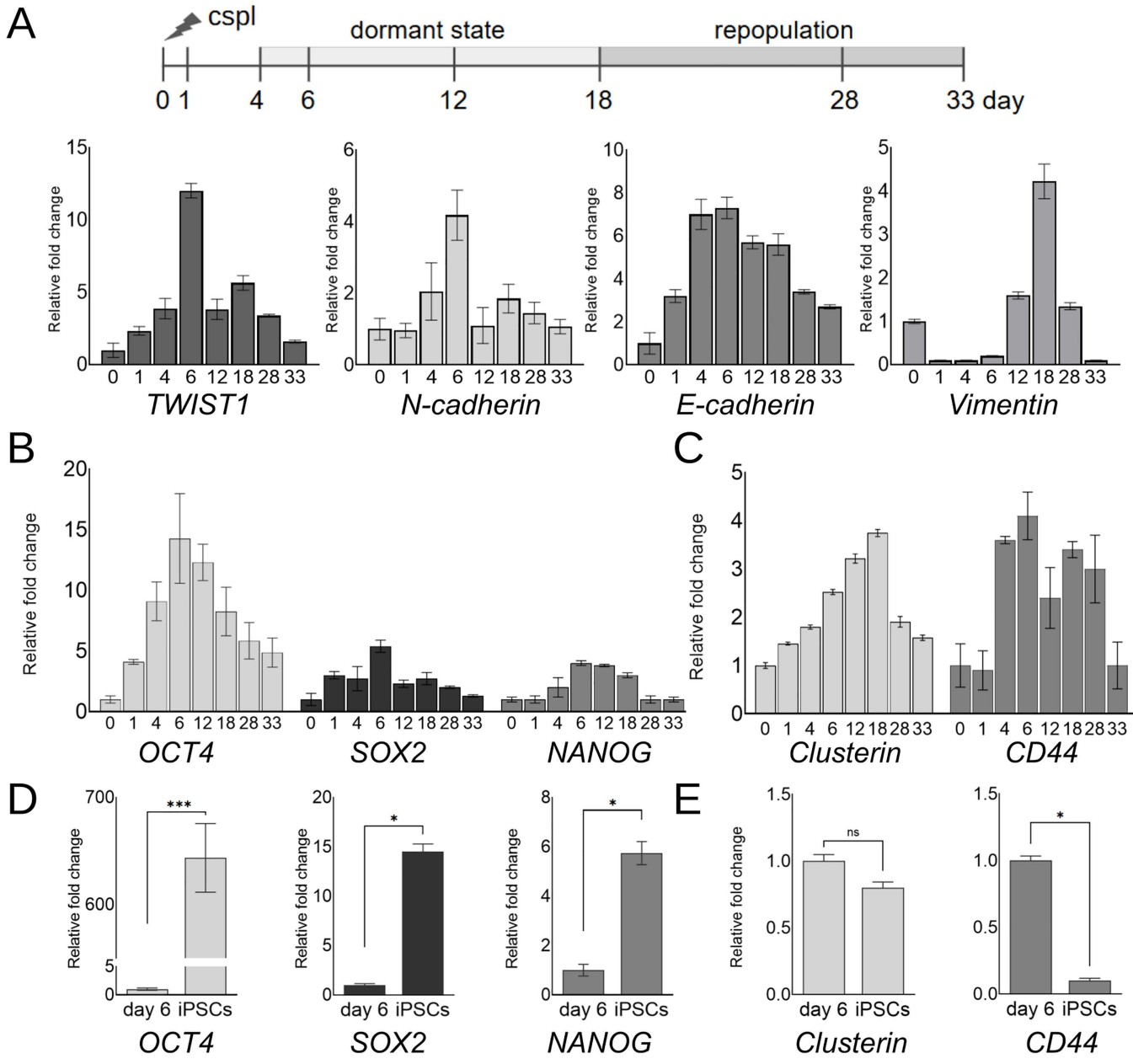

**Fig 6. Markers of EMT and stemness in cancer recurrence model in vitro based on cisplatin-resistant colon cancer cells HCT116 cspl-R, qPCR data. (A)** Normalized expression of genes related to EMT after cisplatin exposure (days 0-33), relative to day 0. **(B,C)** Dynamics of pluripotency markers OCT4, SOX2, NANOG (B) and cancer stem cell markers *clusterin* and CD44 (C) expression, qPCR data. Normalized expression of stemness-related genes after cisplatin exposure (days 0-33), relative to day 0. **(D,E)** Peak activation level of stem cell markers in resistant cells at day 6 after oxaliplatin treatment compared to their expression in human induced pluripotent stem cells (iPSCs) AD3. Expression of the GAPDH gene served as the endogenous control. Data represent biological triplicate experiments and are displayed as mean±SEM.

## Parameter adjustment for cancer recurrence model

In order to adjust the model for its optimal utilisation in recurrence studies, we launched our cancer recurrence model with alternative parameters, including variable oxaliplatin concentrations and time of exposure.

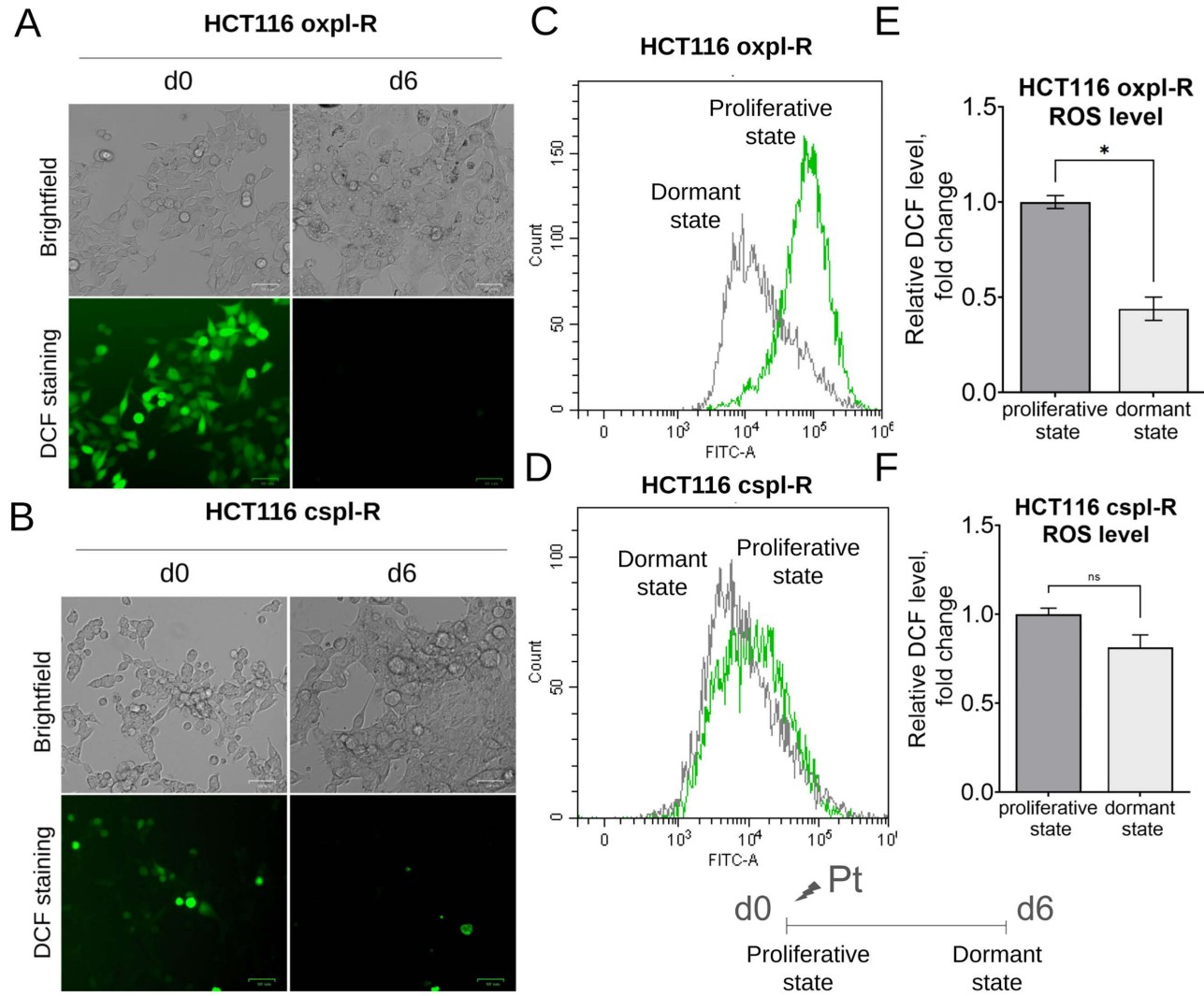

**Fig 7. Differences in ROS levels between proliferative (d0) and quiescent/dormant (d6) HCT116 oxpl-R and HCT116 cspl-R cells.** Representative fluorescent images of DCF-stained HCT116 oxpl-R (A) and HCT116 cspl-R (B) cells. Histograms (C,D) show relative DCF fluorescence levels, demonstrating the distribution of proliferating (green) and dormant (grey) cells according to flow cytometry data. Diagrams (E,F) present mean DCF intensity as relative ROS levels (fold change) based on flow cytometry. Data represent biological triplicate experiments and are displayed as mean±SEM.

For these studies, we used oxaliplatin-resistant cells HCT116 oxpl-R and cisplatin-resistant cells HCT116 cspl-R. Cells were treated with the corresponding platinum drug for 6 or 24 hours, washed, and then cell viability was assessed using the MTT assay every few days until full repopulation (Fig 8). To choose the optimal parameters of the model, HCT116 oxpl-R cells were treated with 50 µM or 150 µM oxaliplatin, while HCT116 cispl-R cells received 25 µM or 75 µM cisplatin. These concentrations are proportional to the known IC50 values and include the working concentrations used for drug resistance development [31,32].

According to the results obtained, in oxaliplatin-resistant cells HCT116 oxpl-R (Fig 8A), short exposure to a high concentration (6h, 150 µM) and prolonged exposure to a moderate concentration (24h, 50 µM) of oxaliplatin showed similar dynamics of cell viability and reached full repopulation by day 18. These two strategies turned out to be the most suitable options for cancer recurrence modeling under these conditions and can be used as alternatives.

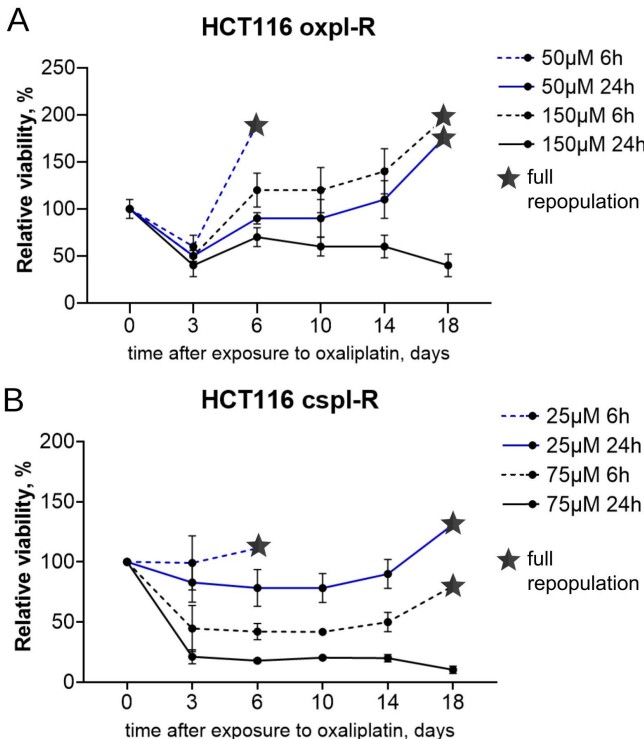

**Fig 8. Testing alternative parameters for cancer recurrence modeling.** Relative viability (MTT assay) of platinum-resistant colon cancer cells: oxaliplatin-resistant HCT116 oxpl-R (A) and cisplatin-resistant HCT116 cspl-R (B) in the cancer recurrence model *in vitro* after different regimens of platinum exposure (50 or 150 µM oxaliplatin/ 25 or 75 µM cisplatin for 6 or 24h). Time points of full repopulation are marked with a star. Data are representative of 6 samples from two independent experiments and are displayed as mean ± SD.

The other two strategies failed to reproduce the model: short 6h exposure to 50 µM oxaliplatin was not enough to block proliferation and induce a dormant state, so repopulation started too soon. Long 24h exposure to 150 µM of oxaliplatin was fatal, and cells did not repopulate at all.

Similarly, in cisplatin-resistant cells HCT116 cspl-R, the same strategies turned out to be valid: long-term exposure to moderate (25 µM) or short-term exposure to high (75 µM) concentration of cisplatin led to repopulation by day 18, while the other two strategies failed to reproduce the model.

Interestingly, for oxaliplatin-resistant cells the key factor is time, so cells after short-term exposure to oxaliplatin (6h) are more viable at both concentrations compared to long-term exposure (24h) and repopulate faster. In contrast, for cisplatin-resistant cells the key factor is the concentration of cisplatin, so cells exposed to lower cisplatin concentration are more viable regardless of the time of exposure. These variations can probably be attributed to distinct mechanisms of platinum resistance operating in these cells.

Obviously, these experiments on adjustment of the model are just broad strokes and can be further refined to fine-tune the model for different purposes.

## Discussion

The *in vitro* model presented in this study recapitulates key features of tumor recurrence, enabling detailed analysis of cancer cell dormancy.

Cancer cell dormancy represents a critical phenomenon in tumor progression, therapy resistance, and disease recurrence. In vitro models of dormancy are essential for studying the mechanisms underlying this quiescent state and developing strategies to target dormant cell populations. The literature reports numerous well-established *in vitro* models designed to investigate the role of various factors in inducing and regulating tumor dormancy [6].

According to Pradhan et al., existing tumor dormancy models can be divided into four major groups based on the type of induction: 1) extracellular matrix (ECM)-induced, 2) cell signaling-induced, 3) biochemical-induced, and 4) drug-induced [6]. While various approaches exist to induce dormancy, drug-induced models are particularly relevant for understanding residual disease following chemotherapy. The tumor recurrence model presented here is drug-induced. The main advantage of this approach lies in its proximity to real clinical conditions, since chemotherapy is one of the primary drivers of drug resistance, dormancy, and tumor recurrence [1].

Drug-induced dormancy reported in literature covers various approaches [6]. Wu et al. induced dormancy in EGF-treated colorectal cancer cells (LoVo, HCT116) using 5-FU for 48 hours, resulting in slow-cycling/dormant cells with elevated stem cell markers (CD133, CD44, LGR5) and enhanced chemoresistance [33]. Li et al. modeled dormancy in breast or prostate cancer cells with high-dose Docetaxel or Doxorubicin, identifying dormant cells after 8–10 days with high p21/Waf1 and PKH26 retention, which resumed proliferation after 18–22 days and formed "recurrent" colonies [34]. Doxorubicin-resistant (DoxR) triple-negative breast cancer (TNBC) models (e.g., MDA-MB-231, BT-549) exhibit dormancy features such as G0/G1 arrest, upregulated p53/CHK2 signaling, and suppressed lipid metabolism. However, these models primarily focus on transient drug tolerance rather than stable resistance with defined genetic mutations, as in our system [35]. Guiro et al. studied dormancy in carboplatin-resistant breast cancer cells using 3D poly(ε-caprolactone) scaffolds, showing scaffold architecture influences dormancy kinetics [36]. Hangauer et al. modeled minimal residual disease in HER2-amplified BT474 cells with lapatinib, producing quiescent survivors that regained proliferation post-drug withdrawal, suggesting non-mutational resistance [37]. In contrast, our model examines persistent resistance from prolonged chemotherapy, likely involving genetic alterations.

Despite numerous models of drug-induced dormancy described to date, models specifically utilizing drug-resistant cancer cells remain scarce in the literature. Drug-resistant cell lines are typically established to investigate mechanisms of resistance, whereas dormancy is often studied either as a related phenomenon or by deriving dormant cells from drug-sensitive lines. In most dormancy-focused studies, drug-sensitive cells undergo short-term treatment, resulting in the simultaneous emergence of resistance and dormancy within the model. The key difference in our approach is the use of drug-resistant cells for modeling.

Major limitations of tumor recurrence modeling include the small size of residual cell populations, lack of an appropriate immune component in recreated niches, and inability to reproduce the real timing of cell reactivation observed in patients.

The first limitation related to dormancy and minimal residual disease models is the typically low yield of surviving dormant cells. Our model addresses this issue by using drug-resistant cells, which enriches the surviving population after cytotoxic exposure and enables high-throughput profiling, functional assays, and bulk cell analyses. While current models focus on minimal residual disease or persistent cells following treatment, their low cell yields restrict the scope of comprehensive studies.

While we cannot directly address the other two limitations related to immune component and recurrence timing in this *in vitro* system, the model's reproducibility and usability of our model make it a promising tool for primary exploratory research. Moreover, the presented model has the potential for further upgrade and refinement, such as incorporating soluble factors into the culture medium or integrating it into direct or indirect co-culture systems with immune cells. Future work will explore adapting this model to *in vivo* conditions, which could help partially overcome the aforementioned limitations by better recapitulating the tumor microenvironment and immune interactions. This advancement would enhance the physiological relevance of the model and expand its applicability for studying tumor dormancy and drug resistance.

We used two platinum-resistant cell lines for tumor recurrence modeling: cisplatin-resistant and oxaliplatin-resistant HCT116 cells. According to our observations, these lines are similar in proliferative activity and morphology. However, in tumor recurrence modeling, these two lines differ in repopulation rates. Repopulation signs appear around day 18 in cisplatin-resistant HCT116 cspl-R cells, with the full repopulation cycle taking 30–40 days. In contrast, oxaliplatin-resistant HCT116 oxpl-R cells repopulate faster, with repopulation beginning around day 12 and completing within 18–20 days. This disparity suggests distinct underlying resistance mechanisms between the two cell lines. Similar variability in repopulation rates was observed by Li et al. in their study modeling dormancy, where human breast (SUM159) and prostate (DU145) tumor cells were subjected to acute docetaxel treatment [34].

An important step was validating the dormant state of cancer cells in our model, according to established dormancy parameters.

Dormancy markers include many diverse characteristics, such as proliferation arrest (G0 phase), high expression of CDK inhibitors with low proliferation marker expression, low ROS levels, a shift in the p-p38:p-ERK kinase ratio toward p-p38, and changes in gene expression regulating cell adhesion, autophagy, programmed cell death, stemness, EMT and other associated markers. [14].

We confirmed cell cycle arrest in platinum-treated cells using gene expression analysis and real-time FUCCI imaging. Yano et al. [38] demonstrated FUCCI as an effective tool to study cancer cell quiescence and identify drugs targeting quiescent cancer cells. A key advantage of using FUCCI in studying residual populations and cancer cell dormancy is the ability to detect cell cycle reactivation in single cells long before visible colonies form. Our FUCCI imaging results correlate well with immunoblotting and qPCR data on the expression of proliferation-related genes. The predominance of FUCCI-red quiescent cells (G0/G1) by the fourth day after platinum treatment of resistant cells corresponds to a clear decrease in the expression of proliferation marker genes KI67 and AURKA, along with accumulation of the cyclin-ependent kinase inhibitor p27/Kip1 (Figs 2,3).

It is established that cancer cell dormancy develops as a result of full suppression of the Raf-MEK-ERK pathway and subsequent G0/G1 proliferation block [39]. Activation of stress pathways, such as p38 signaling, is crucial for dormancy induction [40]. The p38 pathway is activated in dormant metastases [3], whereas growing metastases show increased ERK activity and low p38 activity [39]. We observed a shift in the p-ERK:p-p38 ratio and accumulation of p-p38 during dormancy, persisting until full repopulation. Unexpectedly, p-p38 activity peaked during repopulation rather than dormancy, consistent with reports that exosomal p-p38 can stimulate tumor cell repopulation via p53 regulation [41].

Another important marker linked to the dormant phenotype of cancer cells is the accumulation of stem cell markers, including both markers of cancer stem cells (CSCs) and markers of pluripotency. Three core embryonic stem cell pluripotency regulators, OCT4, SOX2, and NANOG are upregulated in dormant CSCs, contributing to their quiescence and survival [42]. Apart from pluripotency markers, whose activation in dormant cells in our model is less prominent compared to iPSCs (Fig 6), we observed marked upregulation of CD44 and *clusterin*. Stem cell markers such as CD44, CD133, and ALDH1, also known as markers of CSCs, serve as biomarkers for recurrence risk [43]. While certain reports link dormancy to cells lacking CD44 expression (CD44-) [44,45], other studies provide evidence for CD44 upregulation in dormant cells [46,47]. CD44 + cells exhibit enhanced survival and therapy resistance, contributing to cancer dormancy and subsequent cancer recurrence [48]. Clusterin is also expressed in CSCs and linked to poor prognosis and cancer cell survival [49].

Interestingly, CD44 and clusterin promote EMT [50,51], which is involved in dormancy control [20]. Our qPCR data show increased EMT marker expression in the model. (Fig 6A). We showed that resistant cells, after platinum treatment, upregulated the expression of TWIST1, a factor inducing EMT, which allows us to assume that dormant cells in our model undergo EMT. However, epithelial marker E-cadherin, which usually shows opposite dynamics to N-cadherin and is known to be downregulated during EMT, is also upregulated in our model. This unexpected E-cadherin

activation may reflect population heterogeneity or incomplete EMT in dormant cells. It's reported that E-cadherin accumulation in quiescent cells can promote proliferation resumption via mesenchymal-epithelial transition (MET) [20]. There is a concept of epithelial-mesenchymal plasticity (EMP), stating that sometimes cells may not require complete EMT but rather fluid transitions between hybrid epithelial-mesenchymal phenotypes [52,53]. This hybrid state is highly plastic and can switch between different EMT states, enhancing the cancer's ability to invade, metastasize, and resist therapy [26,54].

According to Ruth et al., in dormant residual cells, N-cadherin activation is not always associated with E-cadherin suppression. In particular, in Her2-driven tumors, N-cadherin is more active in dormant cells compared to proliferating tumor cells, while E-cadherin expression remains unchanged. By contrast, in Wnt-driven tumors, E-cadherin levels are elevated in dormant cells without N-cadherin activation [55]. It has also been reported that E-cadherin expression can coexist with the EMT-promoting factor Slug [56].

While our current data do not allow us to identify the exact mechanisms, the findings highlight the complexity of EMT regulation and suggest governing E-cadherin/N-cadherin balance may be critical for understanding their roles in dormant state regulation and therapeutic resistance.

It has been established that dormant or quiescent state is generally accompanied by reduced ROS levels compared to proliferating cells [30]. Although hypoxia-induced dormancy in breast cancer cells may be accompanied by accumulation of ROS and reduced GSH content [57], our findings confirm reduced ROS levels in dormant cells compared to proliferating cells (Fig 7). These conclusions are based on both direct ROS measurement and our results showing the activation of FoxO expression in dormant cells (data not shown), given that FoxO proteins are involved in the control of antioxidant system and have transcriptional targets such as superoxide dismutase-2, peroxiredoxins 3 and 5, and catalase [58].

According to our results, the baseline ROS level in oxaliplatin-resistant HCT116 cells (HCT116 oxpl-R) is significantly higher than in cisplatin-resistant HCT116 cells (HCT116 cspl-R) (Fig 7). Elevated ROS levels are clearly associated with increased tumor malignancy and aggressiveness. For instance, in thyroid cancer, higher ROS generation correlates with more aggressive histological types, advanced tumor stage, and poorer differentiation scores [59]. Similarly, in prostate and breast cancers, increased ROS contribute to enhanced invasive and metastatic phenotypes by promoting oxidative DNA damage and chemoresistance, with ROS acting as signaling molecules that facilitate tumor growth and aggressiveness [60–62]. These findings, together with our data on cross-resistance in two cell lines (S1 Fig), suggest that HCT116 oxpl-R cells represent a more advanced and aggressive platinum-resistant phenotype compared to HCT116 cspl-R cells.

In normal cells, ROS and antioxidants are balanced, but in cancer cells, ROS accumulation disrupts redox homeostasis, promoting genomic instability and proliferation. Chemotherapy-induced ROS overproduction can cause tumor cell death, however, enhanced antioxidant defences in dormant cancer cells enable escape from chemotherapy-induced cell death [63]. In our model, dormant cells showed high resistance to DNA-damaging and antimetabolite drugs compared to proliferating cells (Fig 4). In the model developed by Keeratichamroen et al., dormant A549 cells acquired resistance to several cytotoxic drugs, including etoposide, doxorubicin, paclitaxel and vinblastine [64]. Similarly, breast cancer MDA-MB-231, colon cancer HCT-116, and pancreatic cancer CFPAC cells alter their drug sensitivities, such as to paclitaxel and gemcitabine, upon entering the dormant state [65]. These findings emphasize the urgent need for alternative therapeutic strategies aimed at eliminating quiescent tumor cells, which often evade conventional chemotherapy and contribute to cancer relapse.

## Conclusion

We developed a flexible *in vitro* model based on platinum-resistant colorectal HCT116 cells that reproduces major stages of tumor recurrence, including quiescence/dormancy. Cytotoxic-induced dormancy in our model is characterized by proliferation arrest (G0), low ROS levels, high autophagic activity and other hallmarks consistent with current paradigms of cancer cell dormancy. This model offers broad opportunities for research and serves as a convenient tool to study tumor recurrence and its stages, such as dormancy and repopulation.

## Supporting information

**S1 Fig. Drug resistance and cell cycle analysis in platinum-resistant cell lines.** (A,B) Dose-dependent curves and IC50 tables for (A) cisplatin-resistant (cspl-R), and (B) oxaliplatin-resistant (oxpl-R) HCT116 cells. (C) Colony formation assay comparing oxaliplatin-sensitive HCT116 cells and oxaliplatin-resistant (oxpl-R) cells after 24 h treatment with 5 µM oxaliplatin (day 12). (D) MTT assay showing cross-resistance to platinum drugs in three cell lines: parental HCT116 cells, cisplatin-resistant HCT116 cells (HCT116 cspl-R), and oxaliplatin-resistant HCT116 cells (HCT116 oxpl-R). (E,F) Flow cytometry analysis of DNA content distribution (PI staining) in cisplatin-resistant (HCT116 cspl-R) (E) and oxaliplatin-resistant (HCT116 oxpl-R) (F) cells compared to parental HCT116 cells. Bars depict the percentage of cells with subdiploid DNA. All data represent biological triplicate experiments, mean ± SEM.
(TIF)

**S2 Fig. Uncropped microscopic images of cisplatin-resistant cells HCT116 (HCT116 cspl-R) in 1–33 days after exposure to cisplatin.** The scale bars represent 100 µm.
(TIF)

**S3 Fig. Dynamics of gene expression in the in vitro cancer recurrence model based on oxaliplatin-resistant colon cancer cells HCT116 oxpl-R, analyzed by immunoblotting (A,B) and qPCR (C,D,E).** Immunoblot analysis of Cyclin A, p-ERK, p-p38, LC3, Beclin (A), and Survivin (B) protein levels (α-Tubulin loading control) in oxaliplatin-resistant HCT116 cells (HCT116 oxpl-R) after oxaliplatin exposure (days 0–21). Numbers indicate normalized densitometry values relative to day 0. Normalized expression of proliferation-related genes (AURKA, KI67, Cyclin A, Cycline B, p21/Waf1) (C,D) and stemness-related genes (OCT4, SOX2, NANOG) (E) after oxaliplatin exposure (days 0–22), relative to day 0. Expression of the GAPDH gene served as the endogenous control. Data represent biological triplicate experiments and are displayed as mean ± SEM.
(TIF)

**S1 File. Raw blot images, full-size microphotographs, and the minimal data set can be found in the supporting information file.**
(PDF)

## Author contributions

**Conceptualization:** Maria Igotti.

**Funding acquisition:** Maria Igotti.

**Investigation:** Alisa Morshneva, Olga Gnedina.

**Project administration:** Maria Igotti.

**Supervision:** Maria Igotti.

**Visualization:** Alisa Morshneva.

**Writing – original draft:** Alisa Morshneva.

**Writing – review & editing:** Maria Igotti.

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
