## [Decision Letter · Decision Letter 0]

22 May 2025

Dear Dr. Morshneva,

Thank you for submitting your manuscript to PLOS ONE. After careful consideration, we feel that it has merit but does not fully meet PLOS ONE’s publication criteria as it currently stands. Therefore, we invite you to submit a revised version of the manuscript that addresses the points raised during the review process.

We look forward to receiving your revised manuscript.

Kind regards,

Li Yang, M.D.

Academic Editor

PLOS ONE

Journal Requirements:

“This study was supported by the Russian Science Foundation (project No. 24-25-20164, https://rscf.ru/project/24-25-20164/) and the St. Petersburg Science Foundation (Russia).”

Reviewers' comments:

Reviewer's Responses to Questions

**Comments to the Author**

1. Is the manuscript technically sound, and do the data support the conclusions?

Reviewer #1: Partly

Reviewer #2: Yes

Reviewer #3: Partly

Reviewer #4: Yes

2. Has the statistical analysis been performed appropriately and rigorously?

Reviewer #1: I Don't Know

Reviewer #2: Yes

Reviewer #3: No

Reviewer #4: Yes

3. Have the authors made all data underlying the findings in their manuscript fully available?

Reviewer #1: Yes

Reviewer #2: Yes

Reviewer #3: Yes

Reviewer #4: Yes

4. Is the manuscript presented in an intelligible fashion and written in standard English?

Reviewer #1: No

Reviewer #2: No

Reviewer #3: Yes

Reviewer #4: Yes

Reviewer #1: Dr. Morshneva et al. submitted the manuscript entitled: An in vitro tumor recurrence model based on platinum-resistant colon cancer cells as a research tool for studying cancer cell dormancy, in which the authors reported the establishment of platinum-resistant tumor recurrence model, as well as detailed validation and exploration of the drug-resistant cell line. The authors checked expression level of essential proteins and RNAs as well as ROS level. Generally speaking, this is a meaningful work, and the topic will be of interest to the potential readers of Plos One. However, as characterization and validation are fundamental for reports on a new cell model, I have some major comments for authors to further improvement.

1. The authors did not directly compare drug-resistance cells with parental cells, such as IC50 shifts, survival curves or dose response study.

2. The authors did not perform short-term stability studies of the new cells, such as colony formation study and apoptosis studies when applied to platinum drugs.

3. The authors did not perform long-term stability studies of the new cells, such as passage-stability testing.

4. I’m wondering if the authors observed cross drug-resistance on HCT116 cspl-R or HCT116 oxpl-R? Can these cell lines resist other platinum drugs?

5. The language used occasionally feels a bit choppy, with ideas jumping abruptly from one paragraph to the next.

Considering the importance of this work, but lacking results from some essential experiments, I recommend rejecting the submission and offering the opportunity to resubmit after substantial revisions.

Reviewer #2: This manuscript presents a well-structured and detailed study of a novel in vitro model of tumor recurrence using platinum-resistant HCT116 colon cancer cells. The model aims to recapitulate cellular dormancy and recurrence following oxaliplatin/cisplatin treatment. The authors comprehensively characterize dormancy-associated features, including cell cycle arrest, reduced ROS, increased autophagy, EMT marker expression, and upregulation of stemness-related genes.

The study is methodologically sound and offers a potentially valuable tool for investigating cancer cell dormancy and recurrence. However, several areas would benefit from clarification, more rigorous analysis, and contextualization within the broader literature:

1. The authors state: "To date, no dormancy models utilizing drug-resistant cells have been reported in the literature." (Lines 439–441)

While this claim may be valid, it would benefit from clearer positioning against existing models of dormancy. For example, the recent review by Kamat et al. (Molecular Cancer, 2025, Vol. 24) discusses a range of drug-tolerant and dormancy-related systems. The authors should acknowledge these works and better articulate the specific novelty of their approach.

2. The use of FUCCI labeling is appropriate and informative. However, additional validation—such as Ki-67 immunostaining—would help more definitively distinguish true G0 quiescence from G1 arrest.

3. The manuscript reports dynamic changes in dormancy-related markers (e.g., FoxO1/3, LC3, Beclin, Twist1), but does not explore their causality. Are these markers drivers of dormancy, or merely consequences of treatment-induced stress?

4. The manuscript requires thorough language editing for grammar, clarity, and consistency.

Reviewer #3: General Assessment

In this manuscript (PONE-D-25-19582), Morshneva et al. present a well-conceived and methodologically rigorous in vitro model that simulates tumor recurrence and cancer cell dormancy using platinum-resistant HCT116 colon cancer cells. By exploiting the chemoresistant phenotype, the authors recreate a clinically relevant scenario in which residual, therapy-surviving tumor cells enter a non-proliferative state before eventually repopulating, a key feature of tumor relapse. This approach addresses a fundamental challenge in dormancy research: the limited availability of residual cells post-chemotherapy in traditional models. The model’s high reproducibility and scalability enhance its potential utility in mechanistic studies and preclinical screening of dormancy-targeting therapies.

A key strength of this work lies in the comprehensive and multi-dimensional characterization of the quiescent/dormant state. Through the integration of FUCCI-based cell cycle imaging, qPCR, immunoblotting, ROS quantification, and viability assays, the authors convincingly demonstrate that treated cells enter a reversible growth-arrested state marked by G0/G1 accumulation, reduced ROS levels, increased autophagy, drug resistance, and elevated expression of stemness and EMT markers. The flexibility of the model, demonstrated by manipulating drug type, dose, and exposure time, further enhances its applicability. However, despite the convincing phenotypic profiling, the manuscript would benefit from additional functional validation experiments, clearer handling of interpretive inconsistencies (particularly regarding EMT and treatment comparisons), and improvements in data presentation. Addressing the following concerns is essential for strengthening the manuscript’s scientific rigor and suitability for publication.

Major Comments

1. Inconsistent Use of Resistant Cell Lines: Although both cisplatin- and oxaliplatin-resistant cell lines are described, the authors alternate between them across experiments despite noting that they exhibit different repopulation kinetics (lines 453–458). This inconsistency may raise concerns about selective data presentation. To enhance the model’s generalizability and scientific transparency, key findings should be validated in both cell lines.

2. Lack of Functional Validation of Dormancy Pathways: While dormancy-associated features are well-characterized, functional testing of core pathways (e.g., p38 MAPK, autophagy, ROS detoxification) is lacking. Perturbing these pathways using small-molecule inhibitors (e.g., SB203580 for p38, 3-MA for autophagy) or gene knockdown approaches (e.g., FOXO3 or Beclin1) would significantly strengthen the mechanistic conclusions.

3. Inconsistencies in EMT Marker Expression: The simultaneous upregulation of E-cadherin and N-cadherin is acknowledged but insufficiently addressed. This contradiction could reflect epithelial-mesenchymal plasticity or cellular heterogeneity. Follow-up experiments such as single-cell transcriptomics or immunofluorescence staining are recommended to clarify the phenotype

4. Discussion of Model Limitations: The current discussion does not adequately address the model’s limitations, particularly the absence of tumor microenvironmental components (immune, stromal, and hypoxic factors), which are critical regulators of dormancy in vivo. This gap should be more thoroughly considered in the context of the model’s translational relevance.

5. Data Presentation and Clarity: Several figures and legends are overly dense or lack clarity. Normalization methods, time points, and statistical annotations (including p-values and sample size) should be consistently reported. Graphs would benefit from cleaner labeling and clearer presentation of fold-changes relative to key states (e.g., peak quiescence or baseline proliferation).

6. Figure Legends and Abbreviations: Figure legends should be more comprehensive and include all abbreviations and treatment details referenced in the figure. This will improve interpretability without needing to refer back to the main text.

7. Discrepancy Between Imaging and Graphs in Figure 2: There is a mismatch between the days shown in fluorescent images (days 2 and 4) and the time points in the graph (days 3, 4, 5). This should be clarified or aligned for consistency.

8. Drug Treatment Justification: In Materials and Methods (line 92), the rationale for selecting 25–50 μM cisplatin and 50–150 μM oxaliplatin with exposure times of 6/24 hours should be explicitly stated or supported by literature references.

9. Drug Concentration Justification (Line 276): Drug concentrations for viability testing (e.g., etoposide, irinotecan, 5-FU) are listed without justification. Appropriate references or pilot data should be provided to validate the chosen doses.

10. Inconsistency in Figure 5 Treatments: The rationale for using cisplatin in Figure 5a and oxaliplatin in Figures 5b/c is unclear. Uniform treatment or explanation of the differential use is necessary.

11. Unjustified Reagent Choice (Line 225): The detailed 0–33-day time-course is performed using cisplatin, not oxaliplatin. A justification for this choice is needed, or results should be presented for both agents to support the model’s robustness.

12. Statistical Reporting: The authors report SEM instead of SD without justification. Given the small number of replicates (n=3), reporting SD may more accurately reflect biological variability unless multiple independent experiments were conducted.

13. Misinterpretation of FUCCI Data (Line 224): The claim that cell cycle distribution “returns to its initial state” by day 13 is not fully supported by the data. mKO2-Cdt1-positive cells still represent ~35% versus ~10% at day 0. This needs rephrasing or additional justification.

14. Missing p27 mRNA Data: In Figure 3, the accumulation of p27 is shown by western blot, but not confirmed at the transcript level. Including qPCR results would provide a more complete assessment of cell cycle regulation.

15. Overstated Persistence of Beclin (Line 304): The text claims Beclin accumulation persists through repopulation, but Figure 5 suggests otherwise. This conclusion should be revised or supported with additional quantification.

16. Need for Quantification of Immunoblots (Line 307): The claim that survivin accumulation corresponds with LC3/Beclin decline is not visually convincing. Quantitative densitometry should be provided to support this interpretation.

Minor Comments

• A brief comparison with other drug-induced dormancy models (e.g., breast or lung cancer models treated with doxorubicin or docetaxel) would help contextualize the significance of this work within the field.

• Minor typographical and formatting issues (e.g., inconsistent italicization of gene names, figure label placement) should be corrected throughout the manuscript.

• In Figure 1b, visual cues such as arrows should be added to highlight key features (e.g., vacuolization, multinucleation) for clarity.

Reviewer #4: Overall, this is a clear, concise, and well-written manuscript. The introduction is relevant and research based. Sufficient information about the previous study findings is presented for readers to follow the present study rationale and procedures. The authors make a systematic contribution to the research literature in this area of investigation.

Specific comments follow

p. 4, lines 74: The term "present" suggests that there should be a "presented."

Please provide original full resolution images of Fig 1b

**Do you want your identity to be public for this peer review?** For information about this choice, including consent withdrawal, please see our Privacy Policy

Reviewer #1: No

Reviewer #2: No

Reviewer #3: No

Reviewer #4: **Yes: ** Umar Farooq

---

## [Author Response · Author response to Decision Letter 1]

21 Jul 2025

Response to reviewer 1

We sincerely thank the reviewer for their interest in our manuscript and for the thorough evaluation. Your insightful comments have helped us to improve the quality and informativeness of the article. Below, we will address your questions point by point:

1. The authors did not directly compare drug-resistance cells with parental cells, such as IC50 shifts, survival curves or dose response study.

Since this article is mostly focused on the dormancy and recurrence model, we didn’t include details on developing drug resistance and characterization of the cell lines. HCT116 cspl-R was characterized in a distinct article (10.1134/S1990519X22060037) published earlier, and HCT116 oxpl-R - in another article to be published this year. However, we recognize the importance of this data for dormancy modeling. Therefore, we have included the changes in IC50 as drug resistance develops, along with dose-dependent survival curves for both parental cells and cspl-R/oxpl-R cells, in the Supplementary Materials (S1).

2. The authors did not perform short-term stability studies of the new cells, such as colony formation study and apoptosis studies when applied to platinum drugs.

Regarding the reviewer's comment, we note that a detailed stability characterization of the derived cell lines will be published separately in an upcoming manuscript. However, we have included preliminary data on colony formation, DNA-content distribution using platinum-based compounds in the Supplementary Materials (S1).

3. The authors did not perform long-term stability studies of the new cells, such as passage-stability testing.

Thank you for pointing that out. It should be noted that we do not maintain resistant cells with continuous platinum treatment, as platinum resistance remains stable and persists even after cryopreservation. However, if we observe a decline in resistance, we briefly re-expose the cells to platinum to restore their resistance. This approach ensures consistent resistance levels throughout our experiments. We have clarified this in the Materials and Methods section.

4. I’m wondering if the authors observed cross drug-resistance on HCT116 cspl-R or HCT116 oxpl-R? Can these cell lines resist other platinum drugs?

We appreciate the reviewer’s insightful question regarding cross-resistance in our HCT116 cisplatin-resistant (HCT116 cispl-R) and oxaliplatin-resistant (HCT116 oxpl-R) cell lines. Indeed, we evaluated cross-resistance to other platinum-based drugs (e.g., oxaliplatin in HCT116 cispl-R and cisplatin in HCT116 oxpl-R).

According to our experimental data, cisplatin-resistant HCT116 cspl-R cells do not exhibit cross-resistance to oxaliplatin, a next-generation platinum drug (Fig S1D). However, they show slight cross-resistance to other DNA-damaging agents, including actinomycin D, etoposide, and adriamycin (doxorubicin). Notably, these cells remain sensitive to the topoisomerase I inhibitor irinotecan, the antimetabolite 5-fluorouracil, and histone deacetylase inhibitors such sodium butyrate (data not shown).

Oxaliplatin-resistant cells also demonstrate high resistance to cisplatin, at a level comparable to cisplatin-resistant cells, which allows considering them as a more universal model of generalized resistance to platinum drugs. Nevertheless, the overall spectrum of drugs to which HCT116 oxpl-R and HCT116 cspl-R cells exhibit cross-resistance largely overlaps. We have added this information in the manuscript (data not shown) and provided experimental data on cross-resistance to platinum drugs as supplementary material (Fig S1).

5. The language used occasionally feels a bit choppy, with ideas jumping abruptly from one paragraph to the next.

We have refined the language and incorporated some logical transitions to make the manuscript easier to read.

To sum up our response, we would like to highlight that we have included in the manuscript a brief characterization of the resistant cells, while more comprehensive analysis including karyotyping, transcriptome profiling of oxaliplatin-resistant cells, and cross-resistance data will be published later this year in a separate article.

Response to reviewer 2

Thank you very much for your attention to our study and for your constructive comments. Your input has been valuable in making the article more comprehensive and of higher quality. Below, we will address your questions point by point:

1. The authors state: "To date, no dormancy models utilizing drug-resistant cells have been reported in the literature." (Lines 439–441)

While this claim may be valid, it would benefit from clearer positioning against existing models of dormancy. For example, the recent review by Kamat et al. (Molecular Cancer, 2025, Vol. 24) discusses a range of drug-tolerant and dormancy-related systems. The authors should acknowledge these works and better articulate the specific novelty of their approach.

We appreciate the reviewer’s insightful comment and agree that our statement regarding the novelty of drug-resistant dormancy models could be more precisely contextualized within the existing literature. While we acknowledge the existing of drug-tolerant and dormancy-related systems, our claim specifically refers to the absence of in vitro dormancy models that incorporate drug-resistant cancer cells (e.g., those with defined resistance mutations or long-term resistance phenotypes) to study dormant cell reawakening. After an in-depth literature review, we managed to identify a few models based on resistant cells, and we have cited them in the discussion. To clarify this distinction, we revised the manuscript, referring to a review devoted to modeling cancer cell dormancy, as well as a number of original papers, to take into account broader models of dormancy and drug tolerance: “Drug-induced dormancy reported in literature covers various approaches [6]. Wu et al. induced dormancy in EGF-treated colorectal cancer cells (LoVo, HCT116) using 5-FU for 48 hours, resulting in slow-cycling/dormant cells with elevated stem cell markers (CD133, CD44, LGR5) and enhanced chemoresistance [33]. Li et al. modeled dormancy in breast or prostate cancer cells with high-dose Docetaxel or Doxorubicin, identifying dormant cells after 8-10 days with high p21/Waf1 and PKH26 retention, which resumed proliferation after 18-22 days and formed “recurrent” colonies [34]. Doxorubicin-resistant (DoxR) triple-negative breast cancer (TNBC) models (e.g., MDA-MB-231, BT-549) exhibit dormancy features such as G0/G1 arrest, upregulated p53/CHK2 signaling, and suppressed lipid metabolism. However, these models primarily focus on transient drug tolerance rather than stable resistance with defined genetic mutations, as in our system [35]. Guiro et al. studied dormancy in carboplatin-resistant breast cancer cells using 3D poly(ε-caprolactone) scaffolds, showing scaffold architecture influences dormancy kinetics [36]. Hangauer et al. modeled minimal residual disease in HER2-amplified BT474 cells with lapatinib, producing quiescent survivors that regained proliferation post-drug withdrawal, suggesting non-mutational resistance [37]. In contrast, our model examines persistent resistance from prolonged chemotherapy, likely involving genetic alterations.

Despite numerous models of drug-induced dormancy described to date, models specifically utilizing drug-resistant cancer cells remain scarce in the literature. Drug-resistant cell lines are typically established to investigate mechanisms of resistance, whereas dormancy is often studied either as a related phenomenon or by deriving dormant cells from drug-sensitive lines. In most dormancy-focused studies, drug-sensitive cells undergo short-term treatment, resulting in the simultaneous emergence of resistance and dormancy within the model. The key difference in our approach is the use of drug-resistant cells for modeling.”

2. The use of FUCCI labeling is appropriate and informative. However, additional validation—such as Ki-67 immunostaining—would help more definitively distinguish true G0 quiescence from G1 arrest.

We thank the reviewer for acknowledging the utility of FUCCI labeling in our study and for the suggestion to further validate distinctions between G0 quiescence and G1 arrest. As recommended, we have now performed additional molecular validation using qPCR analysis of ki-67 and key proliferative genes (e.g., aurka, cyclin A, cyclin B). These data have been incorporated into Figure 2D in the revised manuscript.

We have included explanations of the results obtained in the main text of the manuscript:

“qPCR analysis revealed that expression of proliferation markers such as Aurora kinase A (AURKA), MKI67, E2F1, and cyclins A and B significantly decreased 3-6 days after oxaliplatin treatment in both HCT116 oxpl-R and HCT116 cspl-R cell lines (Figs 2D, 3A). This decline coincided with peak mKO2/Cdt1 staining (marking G0/G1 phase) (Fig 2A). Expression level of these markers returns to baseline after repopulation (day 33). Conversely, CDK inhibitors p21/Waf1 and p27/Kip1 accumulate during the quiescent state (days 1–9), followed by downregulation upon cell cycle re-entry (Fig 3B,C). These molecular data provide independent evidence supporting our initial FUCCI-based conclusions regarding cell cycle arrest and recovery.”

Representative Ki-67 immunostaining (or Western blot) was considered. However, qPCR enabled quantitative, multi-gene profiling of cells at different stages of recurrence, aligning best with our mechanistic focus on transcriptomic regulation of quiescence.

We agree that combining dynamic reporters (FUCCI) with molecular markers such as Ki-67 offers a more comprehensive view of cell-cycle states, and believe these additions significantly enhance our findings.

3. The manuscript reports dynamic changes in dormancy-related markers (e.g., FoxO1/3, LC3, Beclin, Twist1), but does not explore their causality. Are these markers drivers of dormancy, or merely consequences of treatment-induced stress?

The observational nature of our data limits causal inferences. FoxO1/3, LC3, Beclin-1, and Twist1 are implicated in dormancy, but their driver or consequence status requires further investigation. These findings highlight their potential as biomarkers and therapeutic targets. Further genetic/pharmacological loss-of-function experiments are needed to determine the role of core pathways or protein in regulation of dormancy entry or maintenance. We intend to present the results of modulating key pathways (p38 MAPK, autophagy, ROS detoxification, FOXO1/3) in dormant cells in a separate publication. These findings may provide new insights into the mechanisms of maintaining the dormant state of cells, which may be useful for developing the therapeutic strategies to prevent relapse, such as reactivating dormant cells for subsequent cytotoxic therapy, maintaining dormancy, or selectively eliminating dormant populations. We are currently conducting experiments to investigate the role of FoxO transcription factors and autophagy modulation in maintaining dormancy using the cell model presented in this Article. This Article is conceptually intended by us as a detailed description of the model, followed by a series of papers presenting experimental results obtained based on the described model, with this article serving as the foundational groundwork.

4. The manuscript requires thorough language editing for grammar, clarity, and consistency.

We have refined the language and incorporated logical transitions to make the manuscript easier to read.

Response to reviewer 3

We appreciate the reviewer’s careful reading of our work and the valuable feedback provided. Your suggestions have significantly strengthened the manuscript and enhanced its overall quality and clarity. Below, we will address your questions point by point:

Major Comments

1. Inconsistent Use of Resistant Cell Lines: Although both cisplatin- and oxaliplatin-resistant cell lines are described, the authors alternate between them across experiments despite noting that they exhibit different repopulation kinetics (lines 453–458). This inconsistency may raise concerns about selective data presentation. To enhance the model’s generalizability and scientific transparency, key findings should be validated in both cell lines.

We appreciate the reviewer’s attention to these details and regret any confusion caused by the data presentation. The discrepancies regarding the cell lines arose from our transition to an oxaliplatin-resistant subline, which was selected for its broader relevance due to cross-resistance across platinum-based therapies. While this subline shares key characteristics with the cisplatin-resistant cells, its enhanced universality better aligns with our study’s goals. To address these concerns directly, we have performed additional validation experiments, including immunoblotting (and qPCR analysis) of key dormancy markers in the oxaliplatin-resistant cells. We expanded the functional characterization of cisplatin-resistant cells with LysoTracker (autophagy) and DCF (ROS) assays to confirm conserved dormancy phenotypes, updating Figures 5 and 7 (and Supplementary Materials) with new experimental data to ensure full transparency.

2. Lack of Functional Validation of Dormancy Pathways: While dormancy-associated features are well-characterized, functional testing of core pathways (e.g., p38 MAPK, autophagy, ROS detoxification) is lacking. Perturbing these pathways using small-molecule inhibitors (e.g., SB203580 for p38, 3-MA for autophagy) or gene knockdown approaches (e.g., FOXO3 or Beclin1) would significantly strengthen the mechanistic conclusions.

We are grateful to the reviewer for this reasonable remark on the development of research directions using the dormancy/tumor recurrence model presented in this Article. We intend to present the results of modulating key pathways (p38 MAPK, autophagy, ROS detoxification, FOXO1/3) in dormant cells in a separate publication. These findings may provide new insights into the mechanisms of maintaining the dormant state of cells, which may be useful for developing the therapeutic strategies to prevent relapse, such as reactivating dormant cells for subsequent cytotoxic therapy, maintaining dormancy, or selectively eliminating dormant populations. We are currently conducting experiments to investigate the role of FoxO transcription factors and autophagy modulation in maintaining dormancy using the cell model presented in this Article. This Article is conceptually intended by us as a detailed description of the model, followed by a series of papers presenting experimental results obtained based on the described model, with this article serving as the foundational groundwork.

3. Inconsistencies in EMT Marker Expression: The simultaneous upregulation of E-cadherin and N-cadherin is acknowledged but insufficiently addressed. This contradiction could reflect epithelial-mesenchymal plasticity or cellular heterogeneity. Follow-up experiments such as single-cell transcriptomics or immunofluorescence staining are recommended to clarify the phenotype

We appreciate the reviewer’s insightful observation regarding the simultaneous upregulation of E-cadherin and N-cadherin in our study. We acknowledge that this apparent contradiction could indeed reflect epithelial-mesenchymal plasticity or cellular heterogeneity within the examined population, as suggested. Clarifying this issue may be an interesting direction for our future work. Thank you for this valuable suggestion, which will guide our further research on this topic. Currently, we analyse transcriptomic data of our oxpl-R cells versus sensitive HCT116 cells. With these data, we can perform a comprehensive study on EMT under oxaliplatin in these cells. While these experiments were beyond the scope of the current manuscript, we have now included a more detailed discussion of this point with additional literature data. We hope this clarification strengthens the interpretation of our findings.

In Text: “Our qPCR data show increased EMT marker expression in the model. (Fig 6A). We showed that resistant cells, after platinum treatment, upregulated the expression of TWIST1, a factor inducing EMT, which allows us to assume th

---

## [Decision Letter · Decision Letter 1]

18 Aug 2025

Dear Dr. Morshneva,

Thank you for submitting your manuscript to PLOS ONE. After careful consideration, we feel that it has merit but does not fully meet PLOS ONE’s publication criteria as it currently stands. Therefore, we invite you to submit a revised version of the manuscript that addresses the points raised during the review process.

We look forward to receiving your revised manuscript.

Kind regards,

Li Yang, M.D.

Academic Editor

PLOS ONE

Journal Requirements:

Reviewers' comments:

Reviewer's Responses to Questions

**Comments to the Author**

Reviewer #1: All comments have been addressed

Reviewer #2: All comments have been addressed

Reviewer #3: All comments have been addressed

Reviewer #4: All comments have been addressed

2. Is the manuscript technically sound, and do the data support the conclusions?

Reviewer #1: Yes

Reviewer #2: Yes

Reviewer #3: Yes

Reviewer #4: No

3. Has the statistical analysis been performed appropriately and rigorously?

Reviewer #1: I Don't Know

Reviewer #2: Yes

Reviewer #3: Yes

Reviewer #4: Yes

4. Have the authors made all data underlying the findings in their manuscript fully available?

Reviewer #1: Yes

Reviewer #2: Yes

Reviewer #3: Yes

Reviewer #4: Yes

5. Is the manuscript presented in an intelligible fashion and written in standard English?

Reviewer #1: Yes

Reviewer #2: Yes

Reviewer #3: Yes

Reviewer #4: Yes

Reviewer #1: (No Response)

Reviewer #2: (No Response)

Reviewer #3: Thank you for carefully addressing my comments and suggestions. The revised manuscript presents the significance of your findings more clearly and effectively. All my previous comments have been addressed, and I appreciate the descriptive and detailed responses provided by the authors, as well as the thoughtful revisions made throughout the manuscript.

Reviewer #4: When applied to platinum medicines, the scientists did not conduct short-term stability investigations of the new cells, such as colony formation and apoptosis studies.

The novel cells were not subjected to long-term stability experiments by the authors, including passage-stability testing.

The figures must be clearly and meaningfully prepared at high quality. Both by themselves and with the typography, the manuscript's images lacked visual appeal. It was suggested that they be changed and modified.

Occasionally, the wording feels a little chaotic, with concepts leaping from one paragraph to the next without explanation.

The article does not investigate the causal relationship between the dynamic changes in dormancy-related markers (such as FoxO1/3, LC3, Beclin, and Twist1) that are reported in the study.

Do these indicators indicate reasons for dormancy, or are they just effects of stress brought on by treatment?

Absence of Functional Validation of Dormancy Pathways: Although characteristics linked to dormancy have been thoroughly described, there is a dearth of functional testing for key pathways (such as p38 MAPK, autophagy, and ROS detoxification). The mechanistic results would be much strengthened if these pathways were perturbed using gene knockdown techniques (e.g., FOXO3 or Beclin1) or smallmolecule inhibitors (e.g., SB203580 for p38, 3-MA for autophagy).

Limitation is missing in the study

Disparities in EMT Marker Expression: Ecadherin and N-cadherin's concurrent overexpression is recognized but not adequately addressed

**Do you want your identity to be public for this peer review?** For information about this choice, including consent withdrawal, please see our Privacy Policy

Reviewer #1: No

Reviewer #2: No

Reviewer #3: No

Reviewer #4: **Yes: ** Dr Umar Farooq

---

## [Author Response · Author response to Decision Letter 2]

26 Aug 2025

Response to reviewer 4

When applied to platinum medicines, the scientists did not conduct short-term stability investigations of the new cells, such as colony formation and apoptosis studies.

Regarding the reviewer's comment, we note that a detailed stability characterization of the derived cell lines will be published separately in an upcoming manuscript. However, we have included preliminary data on colony formation, DNA-content distribution using platinum-based compounds in the Supplementary Materials (S1).

The novel cells were not subjected to long-term stability experiments by the authors, including passage-stability testing.

Thank you for pointing that out. It should be noted that we do not maintain resistant cells with continuous platinum treatment, as platinum resistance remains stable and persists even after cryopreservation. However, if we observe a decline in resistance, we briefly re-expose the cells to platinum to restore their resistance. This approach ensures consistent resistance levels throughout our experiments. We have clarified this in the Materials and Methods section.

The figures must be clearly and meaningfully prepared at high quality. Both by themselves and with the typography, the manuscript's images lacked visual appeal. It was suggested that they be changed and modified.

We improved the quality of the illustrative materials, made the figure legends clearer and more explicit

Occasionally, the wording feels a little chaotic, with concepts leaping from one paragraph to the next without explanation.

We have refined the language and incorporated some logical transitions to make the manuscript easier to read.

The article does not investigate the causal relationship between the dynamic changes in dormancy-related markers (such as FoxO1/3, LC3, Beclin, and Twist1) that are reported in the study. Do these indicators indicate reasons for dormancy, or are they just effects of stress brought on by treatment?

The observational nature of our data limits causal inferences. FoxO1/3, LC3, Beclin-1, and Twist1 are implicated in dormancy, but their driver or consequence status requires further investigation. These findings highlight their potential as biomarkers and therapeutic targets. Further genetic/pharmacological loss-of-function experiments are needed to determine the role of core pathways or protein in regulation of dormancy entry or maintenance. We intend to present the results of modulating key pathways (p38 MAPK, autophagy, ROS detoxification, FOXO1/3) in dormant cells in a separate publication. These findings may provide new insights into the mechanisms of maintaining the dormant state of cells, which may be useful for developing the therapeutic strategies to prevent relapse, such as reactivating dormant cells for subsequent cytotoxic therapy, maintaining dormancy, or selectively eliminating dormant populations. We are currently conducting experiments to investigate the role of FoxO transcription factors and autophagy modulation in maintaining dormancy using the cell model presented in this Article. This Article is conceptually intended by us as a detailed description of the model, followed by a series of papers presenting experimental results obtained based on the described model, with this article serving as the foundational groundwork.

Absence of Functional Validation of Dormancy Pathways: Although characteristics linked to dormancy have been thoroughly described, there is a dearth of functional testing for key pathways (such as p38 MAPK, autophagy, and ROS detoxification). The mechanistic results would be much strengthened if these pathways were perturbed using gene knockdown techniques (e.g., FOXO3 or Beclin1) or smallmolecule inhibitors (e.g., SB203580 for p38, 3-MA for autophagy).

We are grateful to the reviewer for this reasonable remark on the development of research directions using the dormancy/tumor recurrence model presented in this Article. We intend to present the results of modulating key pathways (p38 MAPK, autophagy, ROS detoxification, FOXO1/3) in dormant cells in a separate publication. These findings may provide new insights into the mechanisms of maintaining the dormant state of cells, which may be useful for developing the therapeutic strategies to prevent relapse, such as reactivating dormant cells for subsequent cytotoxic therapy, maintaining dormancy, or selectively eliminating dormant populations. We are currently conducting experiments to investigate the role of FoxO transcription factors and autophagy modulation in maintaining dormancy using the cell model presented in this Article. This Article is conceptually intended by us as a detailed description of the model, followed by a series of papers presenting experimental results obtained based on the described model, with this article serving as the foundational groundwork.

Limitation is missing in the study

In Text: “While we cannot directly address the other two limitations related to immune component and recurrence timing in this in vitro system, the model's reproducibility and usability of our model make it a promising tool for primary exploratory research. Moreover, the presented model has the potential for further upgrade and refinement, such as incorporating soluble factors into the culture medium or integrating it into direct or indirect co-culture systems with immune cells. Future work will explore adapting this model to in vivo conditions, which could help partially overcome the aforementioned limitations by better recapitulating the tumor microenvironment and immune interactions. This advancement would enhance the physiological relevance of the model and expand its applicability for studying tumor dormancy and drug resistance.”

Disparities in EMT Marker Expression: Ecadherin and N-cadherin's concurrent overexpression is recognized but not adequately addressed

We appreciate the reviewer’s insightful observation regarding the simultaneous upregulation of E-cadherin and N-cadherin in our study. We acknowledge that this apparent contradiction could indeed reflect epithelial-mesenchymal plasticity or cellular heterogeneity within the examined population, as suggested. Clarifying this issue may be an interesting direction for our future work. Thank you for this valuable suggestion, which will guide our further research on this topic. Currently, we analyse transcriptomic data of our oxpl-R cells versus sensitive HCT116 cells. With these data, we can perform a comprehensive study on EMT under oxaliplatin in these cells. While these experiments were beyond the scope of the current manuscript, we have now included a more detailed discussion of this point with additional literature data. We hope this clarification strengthens the interpretation of our findings.

In Text: “Our qPCR data show increased EMT marker expression in the model. (Fig 6A). We showed that resistant cells, after platinum treatment, upregulated the expression of TWIST1, a factor inducing EMT, which allows us to assume that dormant cells in our model undergo EMT. However, epithelial marker E-cadherin, which usually shows opposite dynamics to N-cadherin and is known to be downregulated during EMT, is also upregulated in our model. This unexpected E-cadherin activation may reflect population heterogeneity or incomplete EMT in dormant cells. It’s reported that E-cadherin accumulation in quiescent cells can promote proliferation resumption via mesenchymal-epithelial transition (MET) [20]. There is a concept of epithelial-mesenchymal plasticity (EMP), stating that sometimes cells may not require complete EMT but rather fluid transitions between hybrid epithelial-mesenchymal phenotypes [50,51]. This hybrid state is highly plastic and can switch between different EMT states, enhancing the cancer's ability to invade, metastasize, and resist therapy [26,52].

According to Ruth et al., in dormant residual cells, N-cadherin activation is not always associated with E-cadherin suppression. In particular, in Her2-driven tumors, N-cadherin is more active in dormant cells compared to proliferating tumor cells, while E-cadherin expression remains unchanged. By contrast, in Wnt-driven tumors, E-cadherin levels are elevated in dormant cells without N-cadherin activation [53]. It has also been reported that E-cadherin expression can coexist with the EMT-promoting factor Slug [54].

While our current data do not allow us to identify the exact mechanisms, the findings highlight the complexity of EMT regulation and suggest governing E-cadherin/N-cadherin balance may be critical for understanding their roles in dormant state regulation and therapeutic resistance.”

---

## [Decision Letter · Decision Letter 2]

17 Sep 2025

An in vitro tumor recurrence model based on platinum-resistant colon cancer cells as a research tool for studying cancer cell dormancy

PONE-D-25-19582R2

Dear Dr. Morshneva,

We’re pleased to inform you that your manuscript has been judged scientifically suitable for publication and will be formally accepted for publication once it meets all outstanding technical requirements.

Kind regards,

Li Yang, M.D.

Academic Editor

PLOS ONE

Additional Editor Comments (optional):

Thanks for the authors' efforts to comprehensively improve your manuscript according to editor's and reviewers' comments. I am pleased to inform you that your paper can be accepted for publication now. Thanks for the chance to assess your work. Additionally, many thanks for all the reviewers' precious inputs.

Reviewers' comments:

Reviewer's Responses to Questions

**Comments to the Author**

Reviewer #4: All comments have been addressed

2. Is the manuscript technically sound, and do the data support the conclusions?

Reviewer #4: Yes

3. Has the statistical analysis been performed appropriately and rigorously?

Reviewer #4: Yes

4. Have the authors made all data underlying the findings in their manuscript fully available?

Reviewer #4: Yes

5. Is the manuscript presented in an intelligible fashion and written in standard English?

Reviewer #4: Yes

Reviewer #4: Dear Author(s),

I am pleased to inform you that your manuscript has been accepted for publication. The paper makes a valuable contribution to the field of Pharmacy, and the research is well-structured, clearly written, and supported with appropriate methodology and references.

Strengths of the Paper:

The study addresses an important and relevant research problem.

The objectives are clearly stated, and the methodology is sound.

The results are presented effectively with logical interpretation.

The discussion is insightful and supported by recent and relevant literature.

The overall presentation of the manuscript is clear, concise, and professionally written.

I commend the authors for their efforts and the quality of work presented. No further revisions are required at this stage.

Congratulations on your acceptance, and I wish you continued success in your research endeavors.

**Do you want your identity to be public for this peer review?** For information about this choice, including consent withdrawal, please see our Privacy Policy

Reviewer #4: **Yes: ** Dr. Umar Farooq

---

## [Editor Report · Acceptance letter]

PONE-D-25-19582R2

PLOS ONE

Dear Dr. Morshneva,

I'm pleased to inform you that your manuscript has been deemed suitable for publication in PLOS ONE. Congratulations! Your manuscript is now being handed over to our production team.

Kind regards,

on behalf of

Dr. Li Yang

Academic Editor

PLOS ONE